# Detecting Data Deviations in Electronic Health Records

**Kaiping Zheng**[1]**, Horng-Ruey Chua**[2,3]**, Beng Chin Ooi**[4]

[1]School of Computing, National University of Singapore
[2]Division of Nephrology, Department of Medicine, National University Hospital
[3]Department of Medicine, Yong Loo Lin School of Medicine, National University of Singapore
[4]School of Software Technology, Zhejiang University

`dcszkai@nus.edu.sg, mdcchr@nus.edu.sg, ooibc@zju.edu.cn`

## Abstract

Data deviations in electronic health records (EHR) refer to discrepancies between recorded entries and a patient's actual physiological state, indicating a decline in EHR data fidelity. Such deviations can result from pre-analytical variability, documentation errors, or unvalidated data sources. Effectively detecting data deviations is clinically valuable for identifying erroneous records, excluding them from downstream clinical workflows, and informing corrective actions. Despite its importance and practical relevance, this problem remains largely underexplored in existing research. To bridge this gap, we propose a bi-level knowledge distillation approach centered on a task-agnostic formulation of EHR data fidelity as an intrinsic measure of data reliability. Our approach performs layered knowledge distillation in two levels: from a computation-intensive, task-specific data Shapley oracle to a neural oracle for individual tasks, and then to a unified EHR data fidelity predictor. This design enables the integration of task-specific insights into a holistic assessment of a patient's EHR data fidelity from a multi-task perspective. By tracking the outputs of this learned predictor, we detect potential data deviations in EHR data. Experiments on both real-world EHR data from National University Hospital in Singapore and the public MIMIC-III dataset consistently validate the effectiveness of our approach in detecting data deviations in EHR data. Case studies further demonstrate its practical value in identifying clinically meaningful data deviations.

## 1 Introduction

In healthcare data analytics, researchers utilize heterogeneous data sources to support a wide spectrum of applications, including risk prediction, medication recommendation, and disease progression modeling [20, 41]. These efforts have yielded tangible benefits for patients, clinicians, and healthcare institutions, contributing to broader societal impact. Among various data types, electronic health records (EHR) have emerged as a primary source, with increasing availability in recent years. EHR data capture longitudinal patient information, including laboratory results, diagnoses, and prescriptions during their clinical visits. Leveraging EHR data for analytical purposes can enhance patient management and optimize healthcare resource allocation [67, 87].

Despite the potential of EHR data to reveal valuable insights, the aforementioned success of EHR data analytics critically depends on sufficiently high ***data fidelity***, which characterizes how accurately the recorded data capture and reflect the true characteristics of the original EHR data source [89, 79].

However, ensuring high fidelity in EHR data is inherently challenging due to the complexity of clinical environments. Hospitals and healthcare institutions serve large, heterogeneous populations

of patients, each generating numerous medical observations that must be accurately documented for clinical decision-making. The scale and variability of EHR data complicate the maintenance of consistent and reliable records. Common sources of errors include pre-analytical variability during specimen collection and handling [16, 46], as well as documentation errors stemming from human mistakes and complex workflows [80, 3, 14]. The problem is further exacerbated by the integration of non-validated data sources, particularly behavioral and physiological signals from wearable devices [18, 27], which are typically collected in non-clinical, home-based settings and may lack consistency or reliability. These pervasive issues degrade data fidelity and pose substantial barriers to the integrity of EHR-driven analytics and decision support, ultimately constraining the performance of downstream learning models.

Addressing this pressing challenge requires the ability to detect **data deviations** in EHR data—that is, discrepancies between recorded entries in EHR data and the patient's actual physiological state. Detecting such deviations offers two benefits. First, it enables the identification and exclusion of erroneous records from clinical workflows, reducing the risk of inappropriate decisions and adverse outcomes. Second, it facilitates targeted data correction strategies, thereby enhancing data quality and supporting more reliable downstream analysis. More specifically, our primary objective is to enable **pre-hoc detection of potential data deviations at the point of data entry** into the EHR system during clinical practice. The focus is not on post-hoc data cleaning for downstream analytics but on proactive identification of anomalies at the time of data recording. By flagging entries that may exhibit deviations as they are being introduced, we aim to enable real-time quality control and reduce the likelihood of erroneous data entering the system. This proactive mechanism strengthens EHR data quality, which may subsequently enhance the performance of downstream tasks. Nevertheless, this improvement remains a secondary effect; the central goal is the reliable assessment of data fidelity *in situ*, preventing flawed data from affecting clinical decision-making in real-world settings.

Practically significant as it is, detecting data deviations in EHR data remains a non-trivial and underexplored problem. Data deviations directly impair data fidelity, necessitating a principled way to measure data fidelity. The data Shapley value [26], originally derived from the cooperative game theory [69], offers a well-established approach to quantify the contribution of individual data samples to a learning model, making it a promising candidate for this purpose. However, these contribution-based measures—including the data Shapley value and other data valuation techniques [39]—are typically tailored to specific application tasks, and thus may not capture a comprehensive or intrinsic notion of fidelity in a patient's EHR data. In contrast, we aim to develop an application-agnostic fidelity measure that reflects the inherent reliability of a patient's data, independent of downstream tasks, serving as a foundation for more effective detection of data deviations.

**Solution.** We propose an innovative bi-level knowledge distillation approach [33, 28, 62] for detecting EHR data deviations—distinguished by its task-agnostic formulation of data fidelity and its layered transfer of knowledge. As detailed in Section 3, our approach unfolds in three stages. First, we compute task-specific data Shapley values to serve as a data Shapley oracle $\mathcal{O}_{ds}$, encoding fine-grained contribution information for each sample (Section 3.1). Next, to address the high computational cost of $\mathcal{O}_{ds}$, we distill its knowledge into a task-specific neural oracle $\mathcal{O}_{nn}$ by amortizing per-sample computation through a neural network [1, 15], enabling efficient approximation (Section 3.2). Finally, taking a multi-task perspective on data fidelity, we further distill knowledge from all $\mathcal{O}_{nn}$ instances into a unified EHR data fidelity predictor $\Psi$ (Section 3.3). This hierarchical design uniquely enables the integration of task-specific insights into a general-purpose fidelity estimator, which we use to detect and respond to EHR data deviations.

**Novelty.** (i) To the best of our knowledge, this work is the first to address the problem of detecting data deviations in EHR data, a critical yet underexplored issue with direct implications for clinical reliability and safety. (ii) In contrast to existing data valuation methods, such as the data Shapley value, which are inherently task-specific, we propose a multi-task perspective to derive a unified and comprehensive measure of EHR data fidelity. By tracking changes in this task-agnostic fidelity predictor, we enable effective detection of potential deviations in EHR data.

**Contributions.** (i) We address the open problem of measuring EHR data fidelity and detecting data deviations, filling a critical research gap. (ii) We propose a bi-level knowledge distillation approach that transfers information from computationally intensive data Shapley values to task-specific neural oracles, and subsequently to a unified EHR data fidelity predictor, enabling effective and efficient fidelity estimation (Section 3). (iii) We empirically evaluate our approach on both

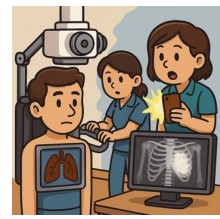 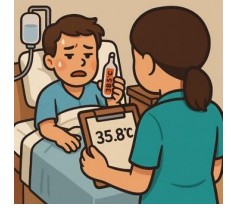 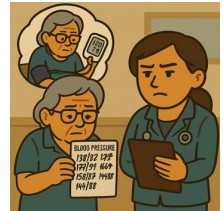

(a) Pre-analytical Variability     (b) Documentation Errors     (c) Non-validated Data Sources

Figure 1: Three representative sources of data deviations in EHR data.

EHR data from National University Hospital in Singapore and the MIMIC-III dataset (Section 4). The results consistently validate the effectiveness of our approach in detecting data deviations, with representative case studies highlighting its practical utility in identifying specific deviation issues relevant to clinical practice.

## 2 Problem and Our Solution

**Necessity of EHR data deviation detection.** As discussed, data deviations in EHR data reflect a decline in the fidelity of recorded health data, raising critical concerns about the extent to which such records accurately represent a patient's physiological state at the time of entry. These deviations may arise from various sources, three of which are illustrated in Figure 1. The first is pre-analytical variability (Figure 1(a)), where deviations are introduced during specimen collection and handling—such as sampling bias, improper storage, or unrecognized technical artifacts. The second is documentation errors (Figure 1(b)), which may occur during data entry, labeling, or integration across systems. These errors, including misclassification, omission, or duplication, typically result from reading mistakes or workflow complexity in interconnected clinical information systems. The third is non-validated data sources (Figure 1(c)), such as patient self-reported measurements, consumer-grade wearable devices, or home-based testing, which lack clinical oversight. These data are prone to inaccuracies due to improper device use, recall bias, or the absence of device calibration and standardization.

For instance, patients admitted to the nephrology department may be required to perform urine tests to assess kidney function. Patients may collect the urine samples at the wrong hours of the day, may contaminate the urine with other body fluids, or healthcare personnel may delay the urine processing; these lead to distorted laboratory results, which may affect subsequent diagnosis and clinical decision-making. In addition, patients with kidney disease and diabetes may have to fast before certain tests for blood glucose, lipids, or parathyroid hormone, for which they may mistakenly consume food prior to testing. In such cases, recall errors or noncompliance can degrade EHR data fidelity, resulting in misleading records and potential data deviations.

The ability to detect potential data deviations in EHR data holds substantial clinical importance. An effective detector can evaluate whether recorded data is trustworthy, thereby improving the accuracy of downstream medical decisions and interventions. Assessing data fidelity allows clinicians to determine whether a given measurement accurately reflects the patient's condition, enabling more precise clinical management. Additionally, identifying patterns of deviations can further guide improvements in data acquisition, collection, and recording protocols, laying the groundwork for future correction and calibration strategies.

**Necessity of distilling knowledge from data Shapley oracle $\mathcal{O}_{\mathrm{ds}}$ to neural oracle $\mathcal{O}_{\mathrm{nn}}$.** To assess EHR data fidelity, the data Shapley value [26]—along with other data valuation methods [39]—offers a promising approach by quantifying the contribution of each data sample. Though inherently tailored to a particular application, the resulting data Shapley values capture the utility of each sample from the perspective defined by the given task, providing a task-specific lens for evaluating data fidelity.

Nevertheless, computing data Shapley values is computationally intensive, with brute-force methods exhibiting exponential complexity. Even with approximation techniques such as Monte Carlo sampling [9], the computational cost remains prohibitive, particularly when applied to large-scale real-world healthcare datasets. This limits the feasibility of applying Shapley-based fidelity assessment in practice, where EHR data fidelity assessment—and thus data deviation detection—requires frequent "what-if" analysis. For example, each newly generated medical feature for a patient may necessitate

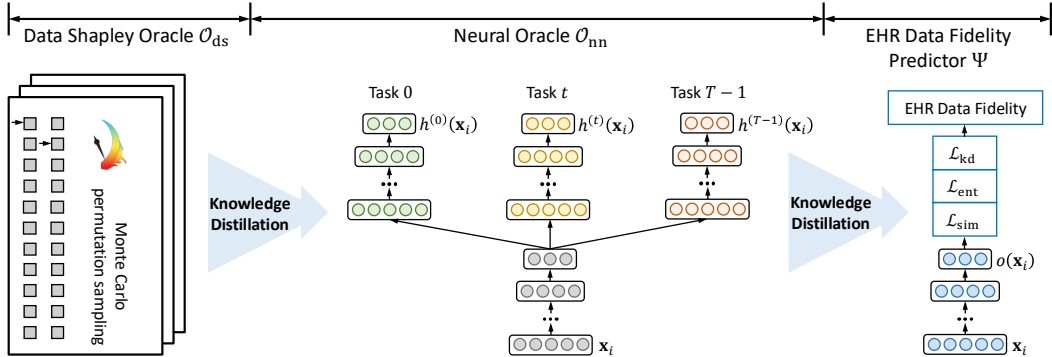

Figure 2: Model architecture of our proposed bi-level knowledge distillation approach.

re-evaluating data fidelity to determine potential deviations. To address this, it is advantageous to amortize the computation across samples using a learned model, such as a neural network [1, 15]. Motivated by this need for efficiency, we propose to distill knowledge from the data Shapley oracle $\mathcal{O}_{\text{ds}}$ into a task-specific neural oracle $\mathcal{O}_{\text{nn}}$, thereby preserving task-relevant valuation insights while significantly reducing computational overhead.

**Necessity of distilling knowledge from neural oracle $\mathcal{O}_{\text{nn}}$ to EHR data fidelity predictor $\Psi$.** We posit that a patient's EHR data fidelity should represent an intrinsic, task-agnostic measure of data reliability. In contrast, the approximate data valuation produced by the task-specific neural oracle $\mathcal{O}_{\text{nn}}$ captures only a partial, task-dependent view. To overcome this limitation, we take a multi-task perspective to distill the task-specific knowledge into a unified EHR data fidelity predictor $\Psi$, which provides a holistic and integrated assessment of a patient's EHR data fidelity by synthesizing insights across multiple clinical tasks, thereby yielding a more comprehensive and robust measure.

## 3 Methodology

Given EHR data $\mathcal{K}^{(t)} = \{k_i^{(t)}\}$ for each task $t$ (i.e., specific application), where $k_i^{(t)}$ denotes the $i$-th sample (i.e., patient in our case), with $i \in \{0, \ldots, N^{(t)} - 1\}$, $t \in \{0, \ldots, T - 1\}$, $N^{(t)}$ is the number of samples for task $t$, and $T$ is the total number of tasks. Each sample is represented as $k_i^{(t)} = (\mathbf{x}_i, y_i^{(t)})$, where $\mathbf{x}_i$ denotes the input EHR data and $y_i^{(t)}$ is the task-specific label. We focus on clinical settings where EHR data and associated labels are available for multiple tasks. Our objective is to first quantify data fidelity—the degree to which a patient's EHR data can be trusted to be accurate and reliable. Based on this, we then detect potential deviations that indicate anomalies or inconsistencies in the EHR data. We define EHR data deviations as follows:

**Definition 1 (EHR Data Deviations)** *Let $\mathbf{x}_i$ denote the original EHR data of patient $i$, and let $\Psi(\mathbf{x}_i)$ represent its data fidelity. Assume $\mathbf{x}_i$ is perturbed to $\mathbf{x}_i + \boldsymbol{\Delta}\mathbf{x}_i$. Given a deviation probability $P_{\text{dev}}$ and a predefined threshold $P_\mu$, a data deviation is detected if the following condition holds:*

$$\exists \delta > 0 \quad such \ that \quad P_{\text{dev}} > P_\mu \quad when \quad \Delta\Psi = \Psi(\mathbf{x}) - \Psi(\mathbf{x} + \boldsymbol{\Delta}\mathbf{x}) > \delta. \tag{1}$$

*In other words, a substantial decline in data fidelity $\Delta\Psi$ beyond a threshold $\delta$ is associated with a high likelihood of deviations in the patient's EHR data.*

To detect EHR data deviations, we propose a bi-level knowledge distillation approach as depicted in Figure 2. In the first level, we train a set of models to approximate the data Shapley values efficiently, one for each prediction task. Concretely, knowledge is distilled from the data Shapley oracle $\mathcal{O}_{\text{ds}}$—which provides application-specific data valuation—into a corresponding neural oracle $\mathcal{O}_{\text{nn}}$ for each task. These task-specific neural oracles are trained jointly in a multi-task learning setting. In the second level, knowledge from all neural oracles is further distilled into a unified EHR data fidelity predictor $\Psi$, which serves to identify potential data deviations. This approach consists of three stages: (i) computing data Shapley values per task via $\mathcal{O}_{\text{ds}}$, (ii) distilling knowledge from $\mathcal{O}_{\text{ds}}$ to $\mathcal{O}_{\text{nn}}$, and (iii) distilling knowledge from $\mathcal{O}_{\text{nn}}$ to $\Psi$. Each stage is detailed below.

## 3.1 Data Shapley Value Computation Per Task in $\mathcal{O}_{\text{ds}}$

We begin by quantifying the value of each patient's EHR data for the available tasks. Data valuation [39] provides a principled framework to assess the contribution of individual data samples to downstream learning performance. Among various approaches, the data Shapley value [26], derived from the Shapley value in cooperative game theory [69], is a widely adopted method with demonstrated effectiveness across multiple domains [65].

Given the task-specific nature of data Shapley values, we compute these values for each sample within each task, resulting in a task-wise data Shapley oracle, denoted as $\mathcal{O}_{\text{ds}}$. For task $t$, let $f^{(t)}$ represent the corresponding prediction model, and consider evaluating its performance on a subset $\mathcal{S} \subseteq \mathcal{K}^{(t)}$. Given an evaluation metric $m$, the performance of $f^{(t)}$ on $\mathcal{S}$ is denoted by $m(\mathcal{S}, f^{(t)})$. The data Shapley value of a specific sample $k_i^{(t)}$ for task $t$ is then defined as:

$$\eta_i^{(t)} = M \sum_{\mathcal{S} \subseteq \mathcal{K}^{(t)} \setminus \{k_i^{(t)}\}} \frac{m(\mathcal{S} \cup \{k_i^{(t)}\}, f^{(t)}) - m(\mathcal{S}, f^{(t)})}{\binom{N^{(t)} - 1}{|\mathcal{S}|}} \tag{2}$$

where the summation considers all subsets of the samples in task $t$, excluding $k_i^{(t)}$, and $M$ is a normalization constant. Equation 2 can be equivalently reformulated as follows:

$$\eta_i^{(t)} = \mathbb{E}_{\varphi \sim \Phi}[m(\mathcal{S}_\varphi^{k_i^{(t)}} \cup \{k_i^{(t)}\}, f^{(t)}) - m(\mathcal{S}_\varphi^{k_i^{(t)}}, f^{(t)})] \tag{3}$$

where $\Phi$ denotes a uniform distribution over all permutations of $\mathcal{K}^{(t)}$, and $\mathcal{S}_\varphi^{k_i^{(t)}}$ is the set of samples preceding $k_i^{(t)}$ in the permutation $\varphi$. Given the exponential complexity of exact Shapley value computation, we approximate $\eta_i^{(t)}$ using Monte Carlo permutation sampling [9] for enhanced efficiency.

## 3.2 Knowledge Distillation from $\mathcal{O}_{\text{ds}}$ to $\mathcal{O}_{\text{nn}}$

Despite the efficiency gains from approximation methods such as Monte Carlo permutation sampling, the computational overhead remains prohibitive, particularly for large-scale real-world healthcare datasets. To address this, we adopt an amortized modeling approach, leveraging neural networks to approximate per-sample outputs and mitigate the cost of individual computations [1, 15].

Concretely, after obtaining task-specific data valuations from the data Shapley oracle $\mathcal{O}_{\text{ds}}$, we distill the knowledge into a neural oracle $\mathcal{O}_{\text{nn}}$ for each task $t$ using a neural network $g^{(t)}(\mathbf{x}, \theta_g^{(t)})$, which is trained to approximate the data Shapley values $\eta_i^{(t)}$ by minimizing the following loss function, where $l(\cdot, \cdot)$ denotes the mean squared error:

$$\mathcal{L}_{\text{ds} \to \text{nn}}^{(t)} = \sum_{k_i^{(t)} \in \mathcal{K}^{(t)}} l(\eta_i^{(t)}, g^{(t)}(\mathbf{x}_i, \theta_g^{(t)})) \tag{4}$$

The neural oracles $\mathcal{O}_{\text{nn}}$ across all tasks are trained jointly in a multi-task learning fashion. To balance task-specific contributions, we assign a weight to each loss $\mathcal{L}_{\text{ds} \to \text{nn}}^{(t)}$ for task $t$, defined as the ratio between its value at the current iteration $s$ and that at the previous iteration $s - 1$:

$$\omega^{(t)} = \mathcal{L}_{\text{ds} \to \text{nn}}^{(t)}(s) / \mathcal{L}_{\text{ds} \to \text{nn}}^{(t)}(s - 1) \tag{5}$$

This weighting strategy encourages loss terms across tasks to decrease at comparable rates, mitigating scale discrepancies and promoting balanced multi-task optimization [30, 50]. We further denote the output of the last hidden layer in $\mathcal{O}_{\text{nn}}$ for task $t$ as $h^{(t)}(\mathbf{x}_i)$, representing the task-specific learned representation of input $\mathbf{x}_i$.

## 3.3 Knowledge Distillation from $\mathcal{O}_{\text{nn}}$ to $\Psi$ for EHR Data Deviation Detection

In this stage, we aim to learn the final EHR data fidelity predictor $\Psi(\mathbf{x}, \theta)$ by distilling knowledge from the task-specific neural oracles $\mathcal{O}_{\text{nn}}$ trained in the previous stage. We construct $\Psi(\mathbf{x}, \theta)$ as a neural network that aggregates and transfers information from all $\mathcal{O}_{\text{nn}}$ across tasks. We denote the hidden representation of $\mathbf{x}$ learned by $\Psi(\mathbf{x}, \theta)$ as $o(\mathbf{x})$.

**Knowledge Distillation Loss.** To ensure that the learned EHR data fidelity predictor $\Psi$ effectively aggregates information from all task-specific neural oracles $\mathcal{O}_{nn}$, we introduce a knowledge distillation loss $\mathcal{L}_{kd}$. This loss minimizes the discrepancy—measured by the mean squared error—between the output of $\Psi(\mathbf{x}_i, \theta)$ and a weighted aggregation of the outputs from the individual $\mathcal{O}_{nn}$ oracles:

$$\mathcal{L}_{kd} = \sum_{\mathbf{x}_i} (\Psi(\mathbf{x}_i, \theta) - \sum_t \alpha^{(t)}(\mathbf{x}_i) g^{(t)}(\mathbf{x}_i, \theta_g^{(t)}))^2 \tag{6}$$

The weight $\alpha^{(t)}(\mathbf{x})$ for task $t$ is computed using an attention subnetwork that integrates both $h^{(t)}(\mathbf{x})$, the task-specific representation of $\mathbf{x}$ from $\mathcal{O}_{nn}$, and $o(\mathbf{x})$, the hidden representation of $o(\mathbf{x})$ derived by $\Psi(\mathbf{x}, \theta)$. The attention subnetwork first concatenates these representations and applies an affine transformation, followed by a ReLU activation:

$$r^{(t)}(\mathbf{x}) = \text{ReLU}(\mathbf{W}_c[o(\mathbf{x}); h^{(t)}(\mathbf{x})] + \mathbf{b}_c) \tag{7}$$

The attention weight $\alpha^{(t)}(\mathbf{x})$ is then computed after normalization across tasks:

$$\tilde{\alpha}^{(t)}(\mathbf{x}) = \mathbf{w}_\alpha^{\mathrm{T}} r^{(t)}(\mathbf{x}) + \mathbf{b}_\alpha, \quad \alpha^{(t)}(\mathbf{x}) = \exp(\tilde{\alpha}^{(t)}(\mathbf{x})) / \sum_{t'} \exp(\tilde{\alpha}^{(t')}(\mathbf{x})) \tag{8}$$

**Relative Entropy Constraint.** In addition to the primary $\mathcal{L}_{kd}$, we introduce a relative entropy-based regularization term to encourage the EHR data fidelity predictor to leverage information from a broader range of neural oracles, rather than over-relying on a small subset. Specifically, we minimize the relative entropy (i.e., Kullback–Leibler divergence) between the learned task-specific attention weights $\alpha^{(t)}(\mathbf{x})$ and a uniform prior $u^{(t)} = 1/T$, thereby encouraging a more balanced utilization of all available neural oracles [76, 60, 59]. This regularization promotes diversity in knowledge aggregation by favoring higher entropy in the attention distribution. The relative entropy constraint is defined as:

$$\mathcal{L}_{ent} = \mathcal{D}_{KL}(\alpha^{(t)}(\mathbf{x}) \| u^{(t)}) = \log T - \mathcal{H}(\alpha^{(t)}(\mathbf{x})) \tag{9}$$

where $\log T$ is a constant, and $\mathcal{H}(\alpha^{(t)}(\mathbf{x}))$ denotes the entropy of $\alpha^{(t)}(\mathbf{x})$ learned from Equation 8.

**Similarity Constraint.** Beyond $\mathcal{L}_{kd}$ and $\mathcal{L}_{ent}$ above, it is also crucial to account for potential redundancy among neural oracles. When two task-specific neural oracles $\mathcal{O}_{nn}$ produce highly similar outputs across samples, assigning high weights to both is unnecessary. To mitigate this, we introduce a similarity constraint to discourage simultaneous high attention weights for such similar oracle pairs:

$$\mathcal{L}_{sim} = \sum_{\mathbf{x}_i} \sum_{t < t'} \alpha^{(t)}(\mathbf{x}_i) \alpha^{(t')}(\mathbf{x}_i) \rho_{t,t'}(\mathbf{x}_i) \tag{10}$$

where $\rho_{t,t'}(\mathbf{x}_i)$ quantifies the output similarity between the neural oracles for tasks $t$ and $t'$ as:

$$\rho_{t,t'}(\mathbf{x}_i) = \exp(-\mathbb{E}[(g^{(t)}(\mathbf{x}_i, \theta_g^{(t)}) - g^{(t')}(\mathbf{x}_i, \theta_g^{(t')}))^2]/\tau) \tag{11}$$

with $\tau$ as a temperature parameter controlling sensitivity [33]. A larger $\rho_{t,t'}(\mathbf{x}_i)$ indicates higher similarity, thus leading to a larger penalty if both tasks are assigned high attention weights.

**Overall Objective.** We integrate the three loss terms—knowledge distillation, entropy regularization, and similarity constraint—into a unified loss function:

$$\mathcal{L} = \lambda_{kd}\mathcal{L}_{kd} + \lambda_{ent}\mathcal{L}_{ent} + \lambda_{sim}\mathcal{L}_{sim} \tag{12}$$

where the weights $\lambda_{kd}$, $\lambda_{ent}$, and $\lambda_{sim}$ are dynamically adjusted based on the ratio of the loss values across successive training iterations:

$$\lambda_{kd} = \mathcal{L}_{kd}(s)/\mathcal{L}_{kd}(s-1), \quad \lambda_{ent} = \mathcal{L}_{ent}(s)/\mathcal{L}_{ent}(s-1), \quad \lambda_{sim} = \mathcal{L}_{sim}(s)/\mathcal{L}_{sim}(s-1) \tag{13}$$

We have thus far described the bi-level knowledge distillation process: first, distilling task-specific data valuation from the data Shapley oracle $\mathcal{O}_{ds}$ into neural oracles $\mathcal{O}_{nn}$, and subsequently distilling this information across tasks into the final EHR data fidelity predictor $\Psi$. The predictor is trained using the overall loss $\mathcal{L}$ and the parameters $\theta$ are optimized iteratively until convergence. Once trained, the EHR data fidelity predictor is used to assess EHR data fidelity and identify potential data deviations, enabling EHR systems to issue alerts when anomalies are detected.

# 4 Experimental Evaluation

We evaluate the effectiveness of our proposed EHR data fidelity predictor $\Psi$ in detecting data deviations using real-world EHR data from National University Hospital in Singapore. To demonstrate its broader applicability, we further assess $\Psi$ on the public MIMIC-III benchmark dataset [40], with results provided in Appendix F.

## 4.1 Experimental Set-up

The evaluation cohort includes patients diagnosed with acute kidney injury (AKI) between November 2015 and October 2016, which in total comprises 2,237 patients from the EHR data of National University Hospital in Singapore. The anonymized dataset contains 43 distinct laboratory tests, resulting in 130,755 recorded test entries. AKI, characterized by a sudden decline in kidney function [4], raises substantial clinical concern regarding long-term prognosis. These patients were subsequently followed up for five years to track clinically significant outcomes. We focus on post-AKI progression by using data from the 90-day period following AKI diagnosis (the "observation window") to predict the occurrence of four major adverse kidney events [68] below during the subsequent 5-year follow-up period (the "prediction window").

- **New or Progressive Chronic Kidney Disease (CKD) Prediction**: Predict whether the patient will experience new-onset or progressive CKD, defined as a decline of more than $30\%$ in baseline estimated glomerular filtration rate (eGFR) after the 90-day observation window.
- **Stage 5 CKD Onset Prediction**: Predict whether the patient's eGFR will decline below $15\mathrm{mL/min/1.73m^2}$ after 90 days, indicating progression to near end-stage kidney disease.
- **Post-AKI Renal Replacement Therapy (RRT) Dependence Prediction**: Predict whether the patient will require RRT during the follow-up period, indicating persistent loss of kidney function requiring long-term intervention.
- **Mortality Prediction**: Predict whether the patient will pass away during the prediction window following the 90-day observation window.

We partition the cohort into $85\%$ for model development and $15\%$ as a held-out set for computing data Shapley values. Within the $85\%$, we further split the data into $80\%$ for training, $10\%$ for validation, and $10\%$ for testing. Hyperparameters are selected based on the best validation performance, measured by the minimum loss $\mathcal{L}$ (Equation 12), across three independent runs. The final model is evaluated on the test set using the selected hyperparameter configuration.

## 4.2 Validation of $\Psi$'s Detection Effectiveness under Controlled Deviation Injection

To evaluate the effectiveness of our proposed EHR data fidelity predictor $\Psi$ in quantifying data fidelity and detecting deviations in EHR data, we design a controlled deviation injection experiment. For each sample in the $10\%$ test set, we identify its most prominent feature, defined as the dimension the value of which is closest to the $99^{\text{th}}$ percentile of the corresponding distribution in the training set, and apply a controlled perturbation. The deviation magnitude is scaled by a multiple of the feature's standard deviation $\sigma$ in the training set. This procedure produces paired samples: the original instance (label $= 0$) and its perturbed counterpart (label $= 1$), forming a labeled benchmark for fidelity evaluation under controlled deviation conditions.

More specifically, this can be conceptually viewed as a "real versus corrupted" classification experiment aimed at distinguishing genuine EHR samples from their perturbed counterparts. Under the controlled deviation injection setup, we (i) perturb each sample by scaling its most prominent feature with its standard deviation $\sigma$ to simulate physiologically plausible deviations, (ii) generate paired instances—original (label $= 0$) and perturbed (label $= 1$), and (iii) construct a labeled benchmark dataset for fidelity evaluation. We then compute the expected EHR data fidelity decline, $\Delta\Psi$, as defined in Equation 1, and use its sign to determine whether the proposed approach successfully detects the introduced deviations.

We compare $\Psi$ against several widely adopted unsupervised anomaly detection baselines:

- **One-Class SVM**: Constructs a decision boundary in a high-dimensional feature space using a radial basis function (RBF) kernel. During training, up to $5\%$ of the samples are allowed to be

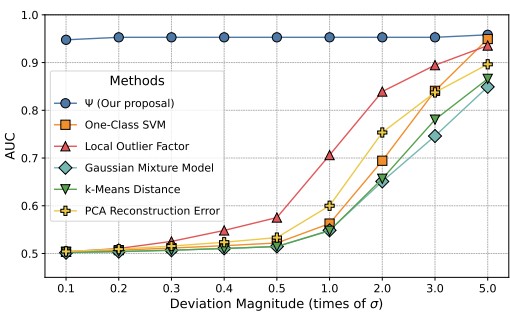 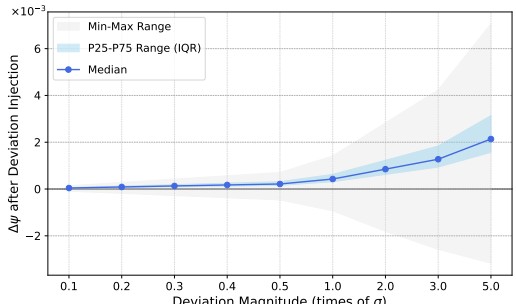

Figure 3: Performance comparison between $\Psi$ and baselines for EHR data deviation detection.

Figure 4: Impact of deviation magnitudes on $\Delta\Psi$ (data fidelity decline) after deviation injection.

treated as outliers. Anomaly scores are derived based on each sample's distance to the learned decision boundary.

- **Local Outlier Factor**: Estimates the local density of each sample based on its 20 nearest neighbors. Samples exhibiting substantially lower local reachability density are identified as outliers. The detection threshold is set at the $5^{th}$ percentile of the Local Outlier Factor scores computed on the training data.
- **Gaussian Mixture Model**: Assumes that the samples are generated from a mixture of Gaussian distributions. Anomaly scores are computed based on the log-likelihood of each sample under the fitted model. Samples with log-likelihoods below the $5^{th}$ percentile of the training distribution are flagged as outliers.
- $k$**-Means Distance**: Computes the Euclidean distance between each sample and its assigned cluster centroid obtained via $k$-Means clustering. Samples with distances exceeding the $90^{th}$ percentile of the training distances are classified as outliers.
- **PCA Reconstruction Error**: Applies principal component analysis (PCA) to reduce dimensionality while retaining $60\%$ of the total variance. Anomaly scores are computed as the reconstruction error for each sample. The $90^{th}$ percentile of reconstruction errors on the training set is used as the detection threshold.

The comparative results in terms of the area under the ROC curve (AUC) are shown in Figure 3. Overall, all evaluated methods exhibit a monotonic increase in AUC as the magnitude of injected deviation grows, which is consistent with intuition: larger perturbations are easier to detect. However, our proposed approach, based on the sign of the fidelity decline ($\Delta\Psi$), demonstrates significantly higher sensitivity to small deviations. Remarkably, even with a perturbation as small as $0.1\sigma$, $\Psi$ detects deviations in the majority of test samples, achieving an AUC of $0.93$. As the deviation magnitude increases, the detection performance improves further, reaching an AUC of $0.95$ at $5\sigma$.

In contrast, baseline methods show little to no response at low perturbation levels. At $0.1\sigma$, their AUC scores remain close to $0.5$, indicating no effective discrimination between perturbed and unperturbed samples. Most baselines only begin to exhibit meaningful detection performance when the deviation exceeds $1\sigma$, and none approach the accuracy of $\Psi$ until the deviation reaches $5\sigma$.

These results highlight the distinct advantage of our approach in detecting subtle yet clinically significant deviations. The fidelity predictor $\Psi$, derived through bi-level knowledge distillation from task-specific data Shapley oracles, provides a powerful signal for identifying deviations in EHR data. In practical clinical settings, deviations of several standard deviations are often already identifiable through conventional rule-based validation. However, it is precisely the small-magnitude deviations— often overlooked yet potentially impactful—that $\Psi$ excels at detecting. This capability enables timely intervention and supports more robust data assurance in real-world healthcare applications.

## 4.3 Validation of $\Psi$'s Output Sensitivity to Varying Deviation Magnitudes

Next, we conduct the output sensitivity experiment using the same paired samples described in Section 4.2, while varying the magnitudes of the injected deviations. Specifically, we examine the fidelity decline, denoted as $\Delta\Psi$, across the perturbed test samples as the deviation magnitude increases from $0.1\sigma$ to $5\sigma$. Figure 4 presents the distribution of $\Delta\Psi$ at each deviation level, reporting

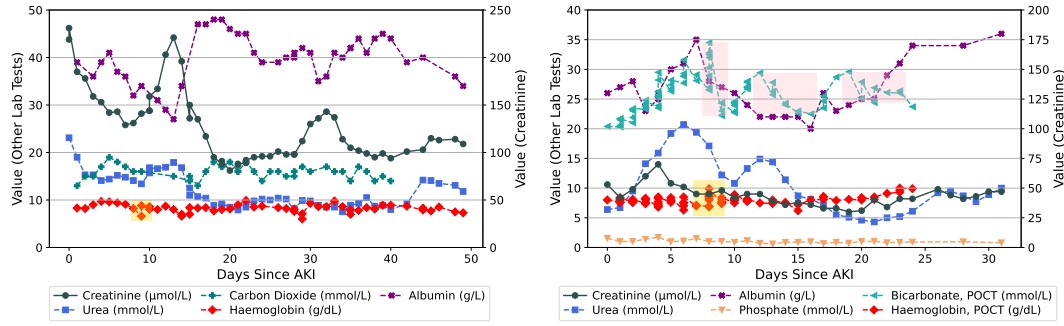

Figure 5: Case study 1.                              Figure 6: Case study 2.

the median, interquartile range (IQR), and full range. As expected, the absolute magnitude of $\Delta\Psi$ increases monotonically with larger deviations, in line with our formulation in Definition 1. Importantly, even when $\Delta\Psi$ is close to zero in absolute terms, its sign remains a reliable signal for identifying deviations. This robustness near the decision boundary is critical, as it ensures $\Psi$'s effectiveness in detecting subtle data shifts without being confounded by noise. Together with the results in Figure 3, this analysis further confirms our approach's strong performance in identifying early-stage deviations that conventional approaches often miss, reinforcing its utility in real-world clinical applications.

### 4.4  Case Study: Detecting Potential Data Deviations in Real-world Scenarios

To further investigate how our proposal could detect potential data deviation issues in real-world scenarios, we zoom in on two example patients from the investigated cohort with low data fidelity values, with their data in the observation window shown.

**Case 1 (Figure 5):** The patient's creatinine level showed a steady decline from 231 μmol/L to 100 μmol/L within a short period, indicating either a marked improvement in renal function or the effect of dialysis to augment renal function [56]. However, the urea level remained elevated—mostly above 10 mmol/L—showing a slight inconsistency with the downward trend of creatinine [35]. Further, in the context of coexisting acidosis (with carbon dioxide persistently below 19 mmol/L [55]) and moderate to severe anemia (with haemoglobin as low as 6.0 g/dL [75]), the serum albumin level paradoxically increased to as high as 48 g/L. This contradicts the fundamental physiological principle that albumin synthesis is typically suppressed under inflammatory and nutritionally imbalanced states [61]. In addition, discrepant duplicate haemoglobin records (e.g., 6.6 and 8.8 g/dL on the same day, highlighted in yellow) were observed, which may suggest the effect of blood transfusion. In a nutshell, the lack of intrinsic physiological consistency among multiple key laboratory tests reflects a lack of pathophysiological coupling between data points, hence suggesting the presence of data deviations in this patient's EHR data. Such records may fail to accurately reflect the patient's actual physiological state at the time and pose a risk of misleading subsequent clinical decisions.

**Case 2 (Figure 6):** The patient's creatinine levels remained consistently within the range of 30–70 μmol/L, corresponding to a nearly normal or only mildly reduced glomerular filtration rate. However, the patient's urea levels remained persistently elevated, with a peak exceeding 20 mmol/L. Such a combination is uncommon [35] in the absence of high protein intake [42] or gastrointestinal bleeding [19], indicating a discordance between the two laboratory test results. Moreover, in the context of hypoalbuminemia (low Albumin [61]), phosphate levels generally below 1.0 mmol/L (with a minimum of 0.6 mmol/L) [58], and development of metabolic acidosis (Bicarbonate, POCT levels [55] repeatedly exhibited sharp drops highlighted in pink), no corresponding metabolic disturbances were observed [44]. Additionally, the patterns of phosphate levels showed no clear relationship with renal function parameters, suggesting the possibility of potential errors or fluctuating nutritional intake. Furthermore, the point-of-care haemoglobin (Haemoglobin, POCT) [57] readings fluctuated markedly within the same day (e.g., from 6.9 to 9.9 g/dL highlighted in yellow). Overall, the biochemical indicators in this patient tend to lack internal physiological consistency, and the dynamic changes in multiple laboratory tests are at odds with the expected pathophysiological characteristics of renal disease, suggesting potential deviations in the recorded data.

## 5 Related Work

Data valuation provides a principled framework to quantify the contribution of individual data samples to the performance of downstream analytic models [39, 65, 73]. Various strategies have been proposed for this purpose. The leave-one-out approach evaluates sample importance by measuring the change in model performance when a sample is excluded from training. Influence functions [43] assess importance based on the model's sensitivity to infinitesimal upweighting of a sample. More recently, the data Shapley value [26], inspired by the Shapley value from cooperative game theory [69], has been introduced as an equitable and theoretically grounded data valuation method. Subsequent work has extended its theoretical foundations [25, 45, 82], improved its computational efficiency [38, 15], and evaluated its practical utility across diverse applications [86, 77].

Amortized computation (or optimization) [1] leverages learning-based models, such as neural networks, to capture shared structure across similar problem instances, enabling efficient prediction of solutions and reducing per-instance computational cost. Compared to non-amortized methods, amortized approaches can offer speedups of several orders of magnitude [1]. This paradigm has been adopted across various domains to improve efficiency, including meta learning [63, 21], explainable machine learning [88, 37], and reinforcement learning [47, 32]. In the context of feature attribution and data valuation, amortized computation is advocated in [15] to address scenarios where exact labels are unavailable or prohibitively costly to obtain. To this end, a stochastic amortization technique is proposed, which trains neural networks using noisy labels while maintaining strong performance with theoretical guarantees.

Knowledge distillation [33, 62, 28] is commonly employed for model compression and acceleration, aiming to transfer knowledge from a large, cumbersome model ("teacher" model) to a smaller, more efficient model ("student" model). The goals of knowledge distillation are multifaceted [36], with two particularly relevant objectives: (i) knowledge compression, where the student model is trained to retain performance comparable to the teacher while being significantly more compact [33, 66, 64]; and (ii) knowledge adaptation, where the student learns to generalize to new or unseen target domains by leveraging knowledge transferred from teacher models trained on related source domains [34, 54].

## 6 Conclusion

This paper addresses the underexplored problem of data deviations in EHR data, which undermines data fidelity in real-world healthcare settings. To detect such deviations, we formulate EHR data fidelity as an intrinsic, task-agnostic property of the data. We then propose a bi-level knowledge distillation approach that transfers knowledge from a task-specific data Shapley oracle ($\mathcal{O}_{ds}$) to a neural oracle ($\mathcal{O}_{nn}$) for each individual task, and subsequently to a unified EHR data fidelity predictor ($\Psi$) that integrates information across tasks. By monitoring the outputs of $\Psi$, our approach enables effective detection of EHR data deviations and, more specifically, supports pre-hoc identification of potential data quality issues at the point of data entry, allowing clinicians to recognize and address erroneous records before they contaminate downstream clinical workflows. Experimental results on the EHR dataset from National University Hospital in Singapore for post-AKI analysis, and on the public MIMIC-III benchmark confirm the effectiveness of the proposed approach. Additionally, representative case studies from the National University Hospital data demonstrate the proposal's ability to pinpoint deviation issues, supporting the identification of erroneous records and guiding correction strategies with practical utility for healthcare practice. Inspired by our case studies, when data fidelity is low or further declines, the detected data deviations may be closely related to complex physiological dynamics in clinical settings. This suggests that additional, uncollected EHR data (even in other data modalities) may need to be gathered and analyzed to fully understand the data deviations and hence the patient's physiological state. This remains an open problem and warrants further in-depth investigation.

## Acknowledgments and Disclosure of Funding

We would like to thank the anonymous reviewers for their constructive comments. NUS' research is partially supported by The Lee Foundation in the form of its NUS Lee Kong Chian Centennial Professorship grant. The electronic AKI data from National University Hospital were supported by the National Kidney Foundation of Singapore Research Grant (NKFRC2014/01/14).

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

# APPENDIX

# A    Notation Table

We adopt the standard convention of using italic symbols (e.g., $x$) for scalars, bold lowercase (e.g., $\mathbf{x}$) for vectors, and bold uppercase (e.g., $\mathbf{X}$) for matrices. Table 1 provides a summary of the notations used throughout the paper. The table is organized into four sections: (i) general notations, (ii) notations for Data Shapley Value Computation Per Task in $\mathcal{O}_{\mathrm{ds}}$ (Section 3.1), (iii) notations for Knowledge Distillation from $\mathcal{O}_{\mathrm{ds}}$ to $\mathcal{O}_{\mathrm{nn}}$ (Section 3.2), and (iv) Knowledge Distillation from $\mathcal{O}_{\mathrm{nn}}$ to $\Psi$ for EHR Data Deviation Detection (Section 3.3).

Table 1: Summary of notations used.

| Notation | Description |
|---|---|
| $\mathcal{O}_{\mathrm{ds}}$ | Data Shapley oracle |
| $\mathcal{O}_{\mathrm{nn}}$ | Task-specific neural oracle |
| $\Psi$ | EHR data fidelity predictor |
| $t$ | Index of a task or specific application |
| $T$ | Total number of tasks |
| $N^{(t)}$ | Number of samples in task $t$ |
| $\mathcal{K}^{(t)}$ | EHR data for task $t$ |
| $k_i^{(t)}$ | The $i$-th EHR sample in task $t$ |
| $\mathbf{x}_i$ | Input features of $k_i^{(t)}$ |
| $y_i^{(t)}$ | Task-specific label of $k_i^{(t)}$ for task $t$ |
| $\Psi(\mathbf{x}_i)$ | Data fidelity of $\mathbf{x}_i$ |
| $\Delta\Psi$ | Decline in data fidelity |
| $f^{(t)}$ | Prediction model used in $\mathcal{O}_{\mathrm{ds}}$ |
| $\mathcal{S}$ | A subset of $\mathcal{K}^{(t)}$ |
| $m$ | Performance evaluation metric |
| $\eta_i^{(t)}$ | Data Shapley value of $k_i^{(t)}$ for task $t$ |
| $\Phi$ | Uniform distribution over all permutations of $\mathcal{K}^{(t)}$ |
| $\varphi$ | A permutation of $\mathcal{K}^{(t)}$ |
| $\mathcal{S}_\varphi^{k_i^{(t)}}$ | Samples preceding $k_i^{(t)}$ in $\varphi$ |
| $g^{(t)}(\mathbf{x}, \theta_g^{(t)})$ | Neural oracle $\mathcal{O}_{\mathrm{nn}}$ for task $t$ |
| $\mathcal{L}_{\mathrm{ds}\to\mathrm{nn}}^{(t)}$ | Knowledge distillation loss from $\mathcal{O}_{\mathrm{ds}}$ to $\mathcal{O}_{\mathrm{nn}}$ for task $t$ |
| $\omega^{(t)}$ | Weight assigned to $\mathcal{L}_{\mathrm{ds}\to\mathrm{nn}}^{(t)}$ |
| $h^{(t)}(\mathbf{x}_i)$ | Learned representation of $\mathbf{x}_i$ in task $t$ |
| $\Psi(\mathbf{x}, \theta)$ | Final EHR data fidelity predictor as a neural network |
| $o(\mathbf{x})$ | Hidden representation of $\mathbf{x}$ learned by $\Psi(\mathbf{x}, \theta)$ |
| $\mathcal{L}_{\mathrm{kd}}$ | Knowledge distillation loss from $\mathcal{O}_{\mathrm{nn}}$ to $\Psi$ |
| $\alpha^{(t)}(\mathbf{x})$ | Attention weight for task $t$ |
| $\mathcal{L}_{\mathrm{ent}}$ | Relative entropy constraint from $\mathcal{O}_{\mathrm{nn}}$ to $\Psi$ |
| $\mathcal{L}_{\mathrm{sim}}$ | Similarity constraint from $\mathcal{O}_{\mathrm{nn}}$ to $\Psi$ |
| $\rho_{t,t'}(\mathbf{x}_i)$ | Output similarity between neural oracles for tasks $t$ and $t'$ |
| $\tau$ | Temperature parameter in similarity computation |
| $\mathcal{L}$ | Overall loss integrating $\mathcal{L}_{\mathrm{kd}}$, $\mathcal{L}_{\mathrm{ent}}$, and $\mathcal{L}_{\mathrm{sim}}$ |
| $\lambda_{\mathrm{kd}}$ | Weight of $\mathcal{L}_{\mathrm{kd}}$ in $\mathcal{L}$ |
| $\lambda_{\mathrm{ent}}$ | Weight of $\mathcal{L}_{\mathrm{ent}}$ in $\mathcal{L}$ |
| $\lambda_{\mathrm{sim}}$ | Weight of $\mathcal{L}_{\mathrm{sim}}$ in $\mathcal{L}$ |

# B    Extended Related Work

## B.1    EHR Data Analytics

In EHR data analytics, heterogeneous EHR data are extensively utilized to support a broad spectrum of applications, including risk prediction, medication recommendation, clinical trial matching, and disease progression modeling [20, 22, 41, 90, 24]. To fully exploit the potential of these valuable data sources, prior studies have investigated multiple methodological directions, such as enhanced representation learning to improve downstream analytic performance [48, 98, 10, 8, 94, 93, 74], cohort modeling to uncover shared patterns across patient subgroups [29, 23, 84, 91, 5, 97, 49], improved explainability and reliability to strengthen clinicians' trust in model outputs [2, 95, 7, 11, 13, 96, 6, 52], and knowledge-guided decision support to integrate domain expertise into predictive modeling [12, 53, 92, 51, 70, 81].

Collectively, these methodological advances enhance patient management and optimize healthcare resource allocation [67, 87], yielding tangible benefits for patients, clinicians, and healthcare institutions. Nevertheless, such progress fundamentally depends on an often implicit assumption—that the underlying EHR data are of high fidelity. This assumption, though critical, is frequently overlooked in real-world clinical environments, posing a major challenge for reliable EHR-based analytics.

## B.2    Data Valuation and Data Shapley Value

Data valuation provides a principled framework to measure the contribution of individual data samples to the performance of downstream analytic models [39, 65, 73]. Several strategies have been developed for this purpose. The leave-one-out approach measures sample importance by evaluating the change in model performance upon excluding each sample. Influence functions [43] estimate importance by analyzing the model's sensitivity to infinitesimal upweighting of a sample.

The data Shapley value [26], inspired by the Shapley value in cooperative game theory [69], has emerged as a theoretically grounded and equitable method for data valuation. Building on this foundation, subsequent works have sought to enhance its theoretical and practical properties. For instance, the distributional Shapley framework [25] generalizes the original formulation by defining the value of each data point over an underlying data distribution. Beta Shapley [45] introduces a relaxation of the efficiency axiom of the Shapley value, which is not essential in machine learning contexts, to achieve desirable statistical properties for efficiency. More recently, a hypothesis testing framework [82] has been presented to examine the data Shapley value under different utility function constraints, motivating a class of utility functions that ensure optimal data selection in such scenarios.

Other researchers seek to reduce the computational overhead associated with the data Shapley value. For example, [38] proposes a suite of techniques to accelerate its computation by introducing specific assumptions on the utility function, enabling practical estimation algorithms for machine learning tasks. In contrast, a unified framework called stochastic amortization [15] is introduced to speed up both feature attribution and data valuation by leveraging amortized computation, which will be discussed in detail in Appendix B.3.

In addition, the practical utility of the data Shapley value has been evaluated in diverse real-world applications. For instance, [86] investigates scenarios where a validation set is unavailable and proposes using the diversity of data samples as an intrinsic property of the dataset, therefore independent of validation. In another study [77], the data Shapley value is employed to quantify the contribution of each training sample to the model's performance in pneumonia detection, using a large chest X-ray medical imaging dataset.

## B.3    Amortized Computation

Amortized computation (or optimization) [1] uses learning-based models, such as neural networks, to exploit shared structure across similar problem instances, thereby enabling efficient solution prediction and significantly reducing per-instance computational cost. Compared to non-amortized methods, amortized approaches can achieve several orders of magnitude in speedup [1], and have been widely adopted to improve efficiency across various domains.

In meta learning, the Model-Agnostic Meta-Learning (MAML) algorithm [21] is designed to be compatible with any model trained via gradient descent, enabling the learning of model parameters that can be easily adaptable. A variant that incorporates implicit differentiation [63] further separates

the computation of the meta-gradient from the specific choice of the inner-loop optimizer, allowing the proposal to tackle a greater number of gradient steps without suffering from vanishing gradients or excessive memory usage.

In explainable machine learning, INVASE [88] performs instance-wise feature selection using a selector-predictor-baseline architecture trained jointly to identify informative subsets of features, where the baseline network is devised to train the selector network. FastSHAP [37] further amortizes the estimation of Shapley values by training an explainer model that approximates them in a single forward pass, using a stochastic gradient descent objective based on weighted least squares.

In reinforcement learning, guided policy search [47] integrates trajectory optimization to direct policy learning, mitigating the risk of poor local optima encountered in direct policy search. Similarly, Stochastic Value Gradients [32] provide a unified framework that leverages backpropagation to learn continuous control policies more effectively.

Finally, in the context of feature attribution and data valuation, amortized computation is employed in [15] to address settings where exact labels are unavailable or expensive to obtain. A stochastic amortization technique is therefore proposed to train neural networks using noisy labels, while still maintaining strong performance backed by theoretical guarantees.

## B.4 Knowledge Distillation

Knowledge distillation [33, 62, 28] is a widely adopted technique for model compression and acceleration. It facilitates the transfer of knowledge from a large, cumbersome model ("teacher" model) to a smaller, more efficient model ("student" model). The objectives of knowledge distillation are multifaceted [36], with two particularly relevant objectives: (i) knowledge compression and (ii) knowledge adaptation, as described below.

In knowledge compression, the goal is to preserve the predictive performance of the teacher model in a significantly more compact student model. In [33], for example, the authors distill the knowledge of an ensemble of models into a single model, achieving competitive performance. In natural language processing, DistilBERT [66] compresses the BERT model via distillation, resulting in improved inference efficiency while preserving core language understanding capabilities. FitNets [64] extend the basic distillation paradigm by training thin and deep student networks using both output predictions and intermediate representations from wide, shallower teacher networks as additional supervision.

In knowledge adaptation, the student model is trained to generalize to new or unseen target domains by leveraging knowledge from teacher models trained on related source domains. For instance, Cycle-Consistent Adversarial Domain Adaptation (CyCADA) [34] enhances domain adaptation by facilitating the alignment in both the generative image space and that in the latent representation space while preserving task-relevant semantics. Another example is the Teacher-Student Curriculum Learning framework [54], where the teacher automatically selects subtasks for the student, enabling progressive learning through a curriculum-based approach.

## C  Pseudocode for Core Stages of the Methodology

This section outlines the pseudocode for the three core stages of the proposed bi-level knowledge distillation approach for detecting data deviations in EHR data. The stages are as follows: (i) computing task-specific data Shapley values using the data Shapley oracle $\mathcal{O}_{\texttt{ds}}$, (ii) distilling knowledge from $\mathcal{O}_{\texttt{ds}}$ to the corresponding task-specific neural oracle $\mathcal{O}_{\texttt{nn}}$, and (iii) distilling knowledge from $\mathcal{O}_{\texttt{nn}}$ to the unified EHR data fidelity predictor $\Psi$.

### C.1  Algorithm 1: Data Shapley Value Computation Per Task in $\mathcal{O}_{\texttt{ds}}$

Algorithm 1 conceptually describes the computation of data Shapley values, which serve as the ground truth supervision for the first level of knowledge distillation.

### C.2  Algorithm 2: Knowledge Distillation from $\mathcal{O}_{\texttt{ds}}$ to $\mathcal{O}_{\texttt{nn}}$

Algorithm 2 details the training procedure for task-specific neural networks $g^{(t)}(\mathbf{x}, \theta_g^{(t)})$, which serve as neural oracles to approximate the data Shapley values produced by $\mathcal{O}_{\texttt{ds}}$.

---

**Algorithm 1** Data Shapley Value Computation Per Task in $\mathcal{O}_{\text{ds}}$

---

**Input:**

$\quad \mathcal{K}^{(t)} = \{(\mathbf{x}_i, y_i^{(t)})\}_{i=0}^{N^{(t)}-1}$: Dataset for task $t \in \{0, \ldots, T-1\}$

$\quad f^{(t)}$: Prediction model for task $t$

$\quad m(\cdot, f^{(t)})$: Performance evaluation metric for task $t$

**Output:**

$\quad \{\eta_i^{(t)}\}_{i=0}^{N^{(t)}-1}$: Data Shapley values for each task $t$

1: **for** each task $t \in \{0, \ldots, T-1\}$ **do**
2: $\quad$ **for** each sample $k_i^{(t)} = (\mathbf{x}_i, y_i^{(t)}) \in \mathcal{K}^{(t)}$ **do**
3: $\quad\quad$ Let $\mathcal{S}_\varphi^{k_i^{(t)}}$ be the set of data samples in $\mathcal{K}^{(t)}$ preceding $k_i^{(t)}$ in a permutation $\varphi$
4: $\quad\quad$ $\eta_i^{(t)} \leftarrow \mathbb{E}_{\varphi \sim \Phi}\left[ m\left(\mathcal{S}_\varphi^{k_i^{(t)}} \cup \{k_i^{(t)}\}, f^{(t)}\right) - m\left(\mathcal{S}_\varphi^{k_i^{(t)}}, f^{(t)}\right)\right]$
5: $\quad$ **end for**
6: **end for**
7: **return** $\{\{\eta_i^{(t)}\}_{i=0}^{N^{(t)}-1}\}_{t=0}^{T-1}$

---

---

**Algorithm 2** Knowledge Distillation from $\mathcal{O}_{\text{ds}}$ to $\mathcal{O}_{\text{nn}}$

---

**Input:**

$\quad \mathcal{D}_{\text{train}}$: Training dataloader yielding batches $(\mathbf{x}_{\text{batch}}, \{\eta_{\text{batch}}^{(t)}\}_{t=0}^{T-1})$

$\quad \mathcal{D}_{\text{val}}$: Validation dataloader

$\quad \{g^{(t)}(\cdot, \theta_g^{(t)})\}_{t=0}^{T-1}$: Set of task-specific neural oracle models

$\quad opt_g$: Optimizer for parameters $\{\theta_g^{(t)}\}$

$\quad \mathcal{L}_{\text{MSE}}$: Mean squared error loss function

$\quad E_1$: Max epochs. $P_1$: Early stopping patience. $\epsilon_1$: Stability constant for weighting

**Output:**

$\quad$ Trained models $\{g^{(t)}(\cdot, \theta_g^{(t)})\}_{t=0}^{T-1}$

1: Initialize $\{\theta_g^{(t)}\}_{t=0}^{T-1}$
2: Initialize $\{\bar{\mathcal{L}}_{\text{prev}}^{(t)} \leftarrow 1.0\}_{t=0}^{T-1}$ (for dynamic per-task loss weighting)
3: **for** epoch $e \leftarrow 1$ **to** $E_1$ **do**
4: $\quad$ **for all** models $g^{(t)}$ **do**
5: $\quad\quad$ $g^{(t)}.\text{train}()$
6: $\quad$ **end for**
7: $\quad$ **for** each batch $(\mathbf{x}_{\text{batch}}, \{\eta_{\text{batch}}^{(t)}\}_{t=0}^{T-1})$ in $\mathcal{D}_{\text{train}}$ **do**
8: $\quad\quad$ $opt_g.\text{zero\_grad}()$
9: $\quad\quad$ $\mathcal{L}_{\text{total\_w}} \leftarrow 0$
10: $\quad\quad$ **for all** tasks $t \in \{0, \ldots, T-1\}$ **do**
11: $\quad\quad\quad$ $\hat{\eta}_{\text{batch}}^{(t)} \leftarrow g^{(t)}(\mathbf{x}_{\text{batch}}, \theta_g^{(t)})$
12: $\quad\quad\quad$ $\mathcal{L}_{\text{batch}}^{(t)} \leftarrow \mathcal{L}_{\text{MSE}}(\eta_{\text{batch}}^{(t)}, \hat{\eta}_{\text{batch}}^{(t)})$
13: $\quad\quad\quad$ Compute dynamic task weight $\omega^{(t)}$:
14: $\quad\quad\quad\quad$ $\omega^{(t)} \leftarrow (e = 1)?1.0 : (\mathcal{L}_{\text{batch}}^{(t)}/(\bar{\mathcal{L}}_{\text{prev}}^{(t)} + \epsilon_1))$
15: $\quad\quad\quad$ $\mathcal{L}_{\text{total\_w}} \leftarrow \mathcal{L}_{\text{total\_w}} + \omega^{(t)} \cdot \mathcal{L}_{\text{batch}}^{(t)}$.
16: $\quad\quad$ **end for**
17: $\quad\quad$ $\mathcal{L}_{\text{total\_w}}.\text{backward}()$
18: $\quad\quad$ $opt_g.\text{step}()$
19: $\quad$ **end for**
20: $\quad$ Update $\{\bar{\mathcal{L}}_{\text{prev}}^{(t)}\}_{t=0}^{T-1}$ with current epoch's computed average task losses
21: $\quad$ Perform validation on $\mathcal{D}_{\text{val}}$; if improvement, save $\{\theta_g^{(t)}\}$; check early stopping ($P_1$)
22: **end for**
23: Load best saved $\{\theta_g^{(t)}\}_{t=0}^{T-1}$
24: **return** $\{g^{(t)}(\cdot, \theta_g^{(t)})\}_{t=0}^{T-1}$

---

## C.3 Algorithm 3: Knowledge Distillation from $\mathcal{O}_{\mathrm{nn}}$ to $\Psi$

Algorithm 3 describes the training of the final predictor $\Psi$ by aggregating knowledge from the task-specific neural oracles $g^{(t)}(\mathbf{x}, \theta_g^{(t)})$ trained in Algorithm 2.

---

**Algorithm 3** Knowledge Distillation from $\mathcal{O}_{\mathrm{nn}}$ to $\Psi$

---

**Input:**
$\quad\mathcal{D}_{\mathrm{train}}$: Training dataloader yielding batches $\mathbf{x}_{\mathrm{batch}}$
$\quad\mathcal{D}_{\mathrm{val}}$: Validation dataloader
$\quad\{g^{(t)}(\cdot, \theta_g^{(t)})\}_{t=0}^{T-1}$: Trained task-specific neural oracles from Algorithm 2 (frozen)
$\quad\Psi(\cdot, \theta_\Psi)$: Unified EHR data fidelity predictor model
$\quad\mathcal{A}(\cdot, \theta_\mathcal{A})$: Attention subnetwork
$\quad opt_\Psi$: Optimizer for $\theta_\Psi, \theta_\mathcal{A}$.
$\quad\tau$: Temperature for $\mathcal{L}_{\mathrm{sim}}$. $T$: Number of tasks
$\quad E_2$: Max epochs. $P_2$: Early stopping patience. $\epsilon_2$: Stability constant for weighting
**Output:**
$\quad$Trained $\Psi(\cdot, \theta_\Psi)$ and $\mathcal{A}(\cdot, \theta_\mathcal{A})$
1: Initialize $\theta_\Psi, \theta_\mathcal{A}$
2: Initialize $\bar{\mathcal{L}}_{\mathrm{prev,kd}}, \bar{\mathcal{L}}_{\mathrm{prev,ent}}, \bar{\mathcal{L}}_{\mathrm{prev,sim}} \leftarrow 1.0$ (for dynamic loss term weighting)
3: **for** epoch $e \leftarrow 1$ **to** $E_2$ **do**
4: $\quad\Psi$.train(); $\mathcal{A}$.train()
5: $\quad$**for** each batch $\mathbf{x}_{\mathrm{batch}}$ **in** $\mathcal{D}_{\mathrm{train}}$ **do**
6: $\quad\quad opt_\Psi$.zero_grad()
7: $\quad\quad$*Perform model forward propagation:*
8: $\quad\quad\quad\{\hat{\eta}_{\mathrm{batch}}^{(t)} \leftarrow g^{(t)}(\mathbf{x}_{\mathrm{batch}}, \theta_g^{(t)})\}_{t=0}^{T-1}$
9: $\quad\quad\quad\{\mathbf{h}_{\mathrm{batch}}^{(t)} \leftarrow g^{(t)}.\mathrm{get\_hidden}(\mathbf{x}_{\mathrm{batch}})\}_{t=0}^{T-1}$
10: $\quad\quad\quad\hat{\Psi}_{\mathrm{batch}} \leftarrow \Psi(\mathbf{x}_{\mathrm{batch}}, \theta_\Psi); \mathbf{o}_{\mathrm{batch}} \leftarrow \Psi.\mathrm{get\_hidden}(\mathbf{x}_{\mathrm{batch}})$
11: $\quad\quad\quad\boldsymbol{\alpha}_{\mathrm{batch}}^{(t)} \leftarrow \mathcal{A}(\mathbf{o}_{\mathrm{batch}} \| \mathbf{h}_{\mathrm{batch}}^{(t)}, \theta_\mathcal{A})$
12: $\quad\quad$*Compute loss terms:*
13: $\quad\quad\quad\mathcal{L}_{\mathrm{kd}} \leftarrow \mathcal{L}_{\mathrm{MSE}}(\hat{\Psi}_{\mathrm{batch}}, \sum_t \boldsymbol{\alpha}_{\mathrm{batch}}^{(t)} \odot \mathrm{detach}(\hat{\eta}_{\mathrm{batch}}^{(t)}))$
14: $\quad\quad\quad\mathcal{L}_{\mathrm{ent}} \leftarrow \mathrm{mean}(\mathcal{D}_{\mathrm{KL}}(\boldsymbol{\alpha}_{\mathrm{batch}} \| \mathrm{Uniform}(1/T)))$
15: $\quad\quad\quad\mathcal{L}_{\mathrm{sim}} \leftarrow \mathrm{mean}_{\mathrm{samples}\ i \in \mathrm{batch}}(\sum_{0 \le t < t' < T} \boldsymbol{\alpha}_i^{(t)} \boldsymbol{\alpha}_i^{(t')} \exp(-\mathrm{MSE}(\hat{\eta}_i^{(t)}, \hat{\eta}_i^{(t')})/\tau))$
16: $\quad\quad$*Compute dynamic weights for loss terms:*
17: $\quad\quad\quad\lambda_{\mathrm{kd}} \leftarrow (e = 1)?1.0 : (\mathcal{L}_{\mathrm{kd}}/(\bar{\mathcal{L}}_{\mathrm{prev,kd}} + \epsilon_2))$; similar for $\lambda_{\mathrm{ent}}, \lambda_{\mathrm{sim}}$
18: $\quad\quad\mathcal{L}_{\mathrm{total}} \leftarrow \lambda_{\mathrm{kd}}\mathcal{L}_{\mathrm{kd}} + \lambda_{\mathrm{ent}}\mathcal{L}_{\mathrm{ent}} + \lambda_{\mathrm{sim}}\mathcal{L}_{\mathrm{sim}}$
19: $\quad\quad\mathcal{L}_{\mathrm{total}}$.backward()
20: $\quad\quad opt_\Psi$.step()
21: $\quad$**end for**
22: $\quad$Update $\bar{\mathcal{L}}_{\mathrm{prev,kd}}, \bar{\mathcal{L}}_{\mathrm{prev,ent}}, \bar{\mathcal{L}}_{\mathrm{prev,sim}}$ with current epoch's computed averages
23: $\quad$Perform validation on $\mathcal{D}_{\mathrm{val}}$; if improvement, save $\theta_\Psi, \theta_\mathcal{A}$; check early stopping ($P_2$)
24: **end for**
25: Load best saved $\theta_\Psi, \theta_\mathcal{A}$
26: **return** $\Psi(\cdot, \theta_\Psi), \mathcal{A}(\cdot, \theta_\mathcal{A})$

---

## D Computational Complexity Analysis

The computational complexity of the three proposed algorithms is analyzed below. The following notations are adopted:

- $T$: Total number of tasks.
- $N^{(t)}$: Number of data samples for task $t$.
- $\mathcal{M}$: Number of Monte Carlo permutations for data Shapley value approximation.
- $N_1$ and $N_2$: Total number of training samples in Algorithm 2 and Algorithm 3, respectively.
- $E_1, E_2$: Number of training epochs for Algorithm 2 and Algorithm 3, respectively.

- $B_1, B_2$: Batch sizes for Algorithm 2 and Algorithm 3, respectively.
- $C_{\text{inf}}^{(t)}$: Computational cost of a single inference (prediction) using the task-specific model $f^{(t)}$ (Algorithm 1).
- $m$: Performance evaluation metric. Evaluating $m(\mathcal{S}, f^{(t)})$ on a subset $\mathcal{S}$ using a fixed model $f^{(t)}$ incurs a cost of $O(|\mathcal{S}| \cdot C_{\text{inf}}^{(t)})$.
- For a generic neural network model $Net$ with parameters $\theta_{Net}$:
    - $C_{\text{fwd}}(Net, \text{batch\_size})$: Cost of a forward pass.
    - $C_{\text{bwd}}(Net, \text{batch\_size})$: Cost of a backward pass (gradient computation), typically approximated as $C_{\text{bwd}} \approx \beta \cdot C_{\text{fwd}}$ for some constant $\beta \geq 1$.
    - $C_{\text{optim}}(Net)$: Cost of updating parameters via an optimizer, typically $O(|\theta_{Net}|)$.
    - $C_{\text{train\_step}}(Net, \text{batch\_size})$: Total cost of a single training step, computed as:
    $$C_{\text{train\_step}}(Net, \text{batch\_size}) = C_{\text{fwd}}(Net, \text{batch\_size}) + C_{\text{bwd}}(Net, \text{batch\_size}) + C_{\text{optim}}(Net)$$
- $g^{(t)}$: Task-specific neural oracle.
- $\Psi$: Unified EHR data fidelity predictor.
- $\mathcal{A}$: Attention subnetwork.

## D.1 Computational Complexity Analysis of Algorithm 1

The exact computation of data Shapley values is known to be #P-hard [72, 17]. To address this, the proposed algorithm adopts a Monte Carlo approximation. For each task $t$ and each of its $N^{(t)}$ samples $k_i^{(t)}$, the data Shapley value is estimated by averaging the marginal contributions across $\mathcal{M}$ random permutations of $\mathcal{K}^{(t)}$.

For a given permutation $\varphi$, the marginal contribution of $k_i^{(t)}$ requires two evaluations of the performance metric $m$: namely, $m(\mathcal{S}_\varphi^{k_i^{(t)}} \cup k_i^{(t)}, f^{(t)})$ and $m(\mathcal{S}_\varphi^{k_i^{(t)}}, f^{(t)})$, where $\mathcal{S}_\varphi^{k_i^{(t)}}$ denotes the set of data points preceding $k_i^{(t)}$ in the permutation, with an expected size of $O(N^{(t)})$. Hence, computing one marginal contribution has a time complexity of $O(N^{(t)} \cdot C_{\text{inf}}^{(t)})$.

Aggregating across all $\mathcal{M}$ permutations and $N^{(t)}$ samples for each task $t$ yields a total complexity of:

$$O\left(\sum_{t=0}^{T-1} \mathcal{M} \cdot N^{(t)} \cdot (N^{(t)} C_{\text{inf}}^{(t)})\right) = O\left(\mathcal{M} \sum_{t=0}^{T-1} (N^{(t)})^2 C_{\text{inf}}^{(t)}\right)$$

Assuming uniform dataset size and inference cost across tasks, i.e., $N^{(t)} \approx N_{\text{avg}}$ and $C_{\text{inf}}^{(t)} \approx C_{\text{inf\_avg}}$, this simplifies to:

$$O(T \cdot \mathcal{M} \cdot N_{\text{avg}}^2 \cdot C_{\text{inf\_avg}})$$

Moreover, under the high-level asymptotic assumption that the per-sample inference cost is constant, i.e., $C_{\text{inf\_avg}} = O(1) + O(\log N_{\text{avg}})$, where the first term arises from the fixed model architecture and input dimensionality, and the second term accounts for AUC computation, the overall computational complexity can be expressed as follows:

$$O(T \cdot \mathcal{M} \cdot N_{\text{avg}}^2 \cdot \log N_{\text{avg}})$$

## D.2 Computational Complexity Analysis of Algorithm 2

In this stage, $T$ task-specific neural oracle models $\{g^{(t)}\}_{t=0}^{T-1}$ are jointly trained for $E_1$ epochs. Each epoch processes $\lceil N_1/B_1 \rceil$ batches, where $N_1$ is the total number of training samples and $B_1$ is the batch size. During each batch, a joint training step is performed using a shared loss function $\mathcal{L}_{\text{total\_w}}$, involving all $T$ models.

Let the per-batch training cost for model $g^{(t)}$ be denoted by $C_{\text{train\_step}}(g^{(t)}, B_1)$. The total training cost across all tasks is given by:

$$O\left(E_1 \cdot \frac{N_1}{B_1} \cdot \sum_{t=0}^{T-1} C_{\text{train\_step}}(g^{(t)}, B_1)\right)$$

Assuming that all task-specific models $g^{(t)}$ have similar computational complexity, we define a representative per-batch training cost as $C_{\text{g\_train\_step}}(B_1)$, yielding:

$$O\left(E_1 \cdot \frac{N_1}{B_1} \cdot T \cdot C_{\text{g\_train\_step}}(B_1)\right)$$

Further assuming that the per-batch training cost scales linearly with the batch size (i.e., $C_{\text{g\_train\_step}}(B_1) = O(B_1)$, with a constant factor determined by a fixed model architecture), the overall time complexity simplifies to:

$$O(E_1 \cdot N_1 \cdot T)$$

### D.3 Computational Complexity Analysis of Algorithm 3

In this algorithm, the unified predictor $\Psi$ and attention subnetwork $\mathcal{A}$ are trained for $E_2$ epochs using $\lceil N_2/B_2 \rceil$ batches. The computational cost per batch consists of the following components:

1. Inference from frozen models: Forward passes through the $T$ frozen task-specific models $g^{(t)}$ to obtain hidden representations $\mathbf{h}^{(t)}$, with total cost $\sum_{t=0}^{T-1} C_{\text{fwd}}(g^{(t)}, B_2)$.

2. $\Psi$ forward pass: Computes the prediction $\hat{\Psi}$ and hidden representation $\mathbf{o}$, with cost $C_{\text{fwd}}(\Psi, B_2)$.

3. $\mathcal{A}$ forward pass: Computes attention scores with cost $C_{\text{fwd}}(\mathcal{A}, B_2)$.

4. Loss computation:
    - Knowledge distillation loss $\mathcal{L}_{\text{kd}}$: $O(B_2 T)$.
    - Relative entropy constraint $\mathcal{L}_{\text{ent}}$: $O(B_2 T)$.
    - Similarity constraint $\mathcal{L}_{\text{sim}}$: $O(B_2 T^2)$ (due to pairwise comparisons).

    The dominant cost among these is:

    $$C_{\text{loss\_calc}} = O(B_2 T^2)$$

5. Backward pass for $\Psi, \mathcal{A}$: $C_{\text{bwd}}(\Psi, B_2) + C_{\text{bwd}}(\mathcal{A}, B_2)$.
6. Optimizer update for $\theta_\Psi, \theta_\mathcal{A}$: $C_{\text{optim}}(\Psi) + C_{\text{optim}}(\mathcal{A})$.

Aggregating these components, the total time is:

$$O\left(E_2 \cdot \frac{N_2}{B_2} \cdot \left[\sum_{t=0}^{T-1} C_{\text{fwd}}(g^{(t)}, B_2) + C_{\text{train\_step}}(\Psi, B_2) + C_{\text{train\_step}}(\mathcal{A}, B_2) + C_{\text{loss\_calc}}\right]\right)$$

Assuming an average forward pass cost $C_{\text{g\_fwd\_avg}}(B_2)$ for each of the $T$ frozen $g^{(t)}$ models, and recalling that $C_{\text{loss\_calc}} = O(B_2 T^2)$, the total time complexity simplifies to:

$$O\left(E_2 \cdot \frac{N_2}{B_2} \cdot \left[T \cdot C_{\text{g\_fwd\_avg}}(B_2) + C_{\text{train\_step}}(\Psi, B_2) + C_{\text{train\_step}}(\mathcal{A}, B_2) + B_2 T^2\right]\right)$$

In the high-level asymptotic case where all forward and training step costs scale linearly with batch size (i.e., $C_{\text{g\_fwd\_avg}}(B_2) = O(B_2)$, $C_{\text{train\_step}}(\Psi, B_2) = O(B_2)$, and $C_{\text{train\_step}}(\mathcal{A}, B_2) = O(B_2)$), the expression becomes:

Table 2: Prevalence per task in the original AKI dataset.

| Task | # Positive | % |
|---|---|---|
| New or Progressive CKD | 941 | 32.58 |
| Stage 5 CKD Onset | 295 | 10.21 |
| Post-AKI RRT Dependence | 118 | 4.09 |
| Mortality | 670 | 23.20 |

$$O\left(E_2 \cdot \frac{N_2}{B_2} \cdot (T \cdot B_2 + B_2 + B_2 + B_2 T^2)\right) = O(E_2 \cdot N_2(T + T^2))$$

Therefore, the most simplified dominant time complexity is:

$$O(E_2 \cdot N_2 \cdot T^2)$$

# E    Extended Experimental Set-up

This section provides additional details of the experimental set-up for both the AKI (acute kidney injury) dataset from National University Hospital in Singapore and the publicly available MIMIC-III benchmark dataset [40].

For both datasets, we partition the extracted samples into $85\%$ for model development and $15\%$ as a held-out set for computing data Shapley values. Within the development set, we further divide the data into $80\%$ for training, $10\%$ for validation, and $10\%$ for testing. Model training is performed using the Adam optimizer. Hyperparameters are selected based on the best validation performance, measured by the minimum loss $\mathcal{L}$ (Equation 12), averaged over three independent runs. The final model is then evaluated on the test set using the selected hyperparameter configuration. Additional setup details specific to each dataset are provided in Appendices E.1 and E.2.

The experiments are conducted on a server equipped with two Intel Xeon Gold 6248R CPUs, 768 GB of memory, and eight NVIDIA V100 GPUs connected via NVLINK. All models are implemented using PyTorch version 1.12.1.

To evaluate the effectiveness of our proposed bi-level knowledge distillation approach, we compute the expected EHR data fidelity decline, $\Delta\Psi$, as defined in Equation 1, and use its sign to indicate the presence or absence of a detected deviation.

## E.1    Experimental Set-up on the AKI Dataset

The original cohort from National University Hospital in Singapore comprises 2,888 patients diagnosed with AKI (acute kidney injury) [4] between November 2015 and October 2016, with follow-up data collected over a five-year period to monitor their post-AKI outcomes. We focus on four major adverse kidney events (MAKE) [68], which serve as four prediction tasks in our setting and reflect long-term deterioration in kidney function. These events include: (i) the development of new or progressive CKD (chronic kidney disease), defined as a decline of more than $30\%$ in baseline eGFR; (ii) the onset of Stage 5 CKD, indicated by eGFR falling below $15\mathrm{mL/min/1.73m^2}$; (iii) dependence on RRT (renal replacement therapy); and (iv) mortality. The prevalence of each task (i.e., proportion of positive samples) in the original AKI dataset is summarized in Table 2.

We define a 90-day observation window following the initial AKI diagnosis and use the patients' laboratory test results within this period as model input. Applying this criterion results in the exclusion of 651 patients (without laboratory tests in the observation window), yielding a final cohort of 2,237 patients. The objective is to predict the occurrence of the four target events within a subsequent prediction window. The temporal relationship between the observation and prediction windows is illustrated in Figure 7. Specifically, we extract 43 distinct types of laboratory tests recorded during the observation window, comprising a total of 130,755 test entries.

Regarding the hyperparameter settings, the task-specific neural oracle $\mathcal{O}_{\mathrm{nn}}$ is implemented as a multilayer perceptron (MLP) with three hidden layers of sizes 32, 16, and 8, respectively. The unified

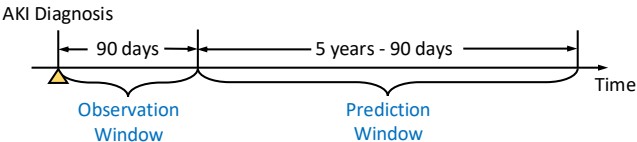

Figure 7: Relationship between observation window and prediction window in the AKI dataset.

Table 3: Prevalence per task in the MIMIC-III dataset.

| Task | # Positive | % |
|---|---|---|
| Essential hypertension | 17573 | 41.94 |
| Coronary atherosclerosis and other heart disease | 13540 | 32.31 |
| Cardiac dysrhythmias | 13458 | 32.12 |
| Disorders of lipid metabolism | 12162 | 29.02 |
| Fluid and electrolyte disorders | 11254 | 26.86 |
| Congestive heart failure; nonhypertensive | 11220 | 26.78 |
| Acute and unspecified renal failure | 8964 | 21.39 |
| Complications of surgical procedures or medical care | 8695 | 20.75 |
| Diabetes mellitus without complication | 8074 | 19.27 |
| Respiratory failure; insufficiency; arrest (adult) | 7566 | 18.06 |
| Septicemia (except in labor) | 5975 | 14.26 |
| Pneumonia (except that caused by tuberculosis or sexually transmitted disease) | 5815 | 13.88 |
| Chronic kidney disease | 5607 | 13.38 |
| Hypertension with complications and secondary hypertension | 5547 | 13.24 |
| Chronic obstructive pulmonary disease and bronchiectasis | 5455 | 13.02 |
| Acute myocardial infarction | 4337 | 10.35 |
| Diabetes mellitus with complications | 3988 | 9.52 |
| Other liver diseases | 3723 | 8.89 |
| Pleurisy; pneumothorax; pulmonary collapse | 3658 | 8.73 |
| Shock | 3291 | 7.85 |
| Acute cerebrovascular disease | 3079 | 7.35 |
| Gastrointestinal hemorrhage | 3067 | 7.32 |
| Conduction disorders | 3011 | 7.19 |
| Other lower respiratory disease | 2168 | 5.17 |
| Other upper respiratory disease | 1702 | 4.06 |

EHR data fidelity predictor $\Psi$ is also an MLP, with hidden layers of sizes 64, 32, and 16. The representation dimension of $r^{(t)}(\mathbf{x})$ in the attention subnetwork (Equation 7) is set to 32. We use a learning rate of 0.01 for training $\mathcal{O}_{nn}$ and 0.0001 for $\Psi$. The temperature parameter $\tau$ in Equation 11 is set to 0.5. Training is conducted for a maximum of 1000 epochs with a batch size of 128. Early stopping is employed if the validation performance does not improve for 50 consecutive epochs.

### E.2 Experimental Set-up on the MIMIC-III Dataset

MIMIC-III [40] (Medical Information Mart for Intensive Care) is a widely used benchmark dataset in EHR data analytics. It comprises EHR data from over forty thousand patients admitted to intensive care units (ICUs) between 2001 and 2012. In this study, we adopt the multi-task learning benchmark established in [31], focusing on the phenotype classification application. This application involves predicting the presence of 25 distinct acute care conditions (i.e., phenotypes) during a given ICU stay, formulated as a multilabel classification problem.

In this dataset, a single patient may have multiple hospital admissions, and each admission can include multiple ICU stays. Following the protocol in [31], we treat each ICU stay as an individual sample, resulting in a total of 41,902 samples. The goal of phenotype classification is to predict the presence of specific acute care conditions (phenotypes) for each ICU stay. Detailed descriptions of the phenotypes and their corresponding prevalences (i.e., the proportion of positive samples) are provided in Table 3.

Table 4: Tuning range for hyperparameters on the AKI dataset.

| Hyperparameters | Tuning Range |
|---|---|
| Dimensions of $g^{(t)}(\mathbf{x}, \theta_g^{(t)})$ | [64, 32, 16], [32, 32, 16], [32, 16, 8] |
| Dimensions of $\Psi(\mathbf{x}, \theta)$ | [128, 64, 32], [64, 32, 16], [32, 16, 8] |
| Dimension of $r^{(t)}(\mathbf{x})$ | {16, 32} |
| Learning rate of $g^{(t)}(\mathbf{x}, \theta_g^{(t)})$ | {0.0001, 0.001, 0.01} |
| Learning rate of $\Psi(\mathbf{x}, \theta)$ | {0.0001, 0.001, 0.01} |
| Temperature $\tau$ | {0.5, 1.0, 2.0} |

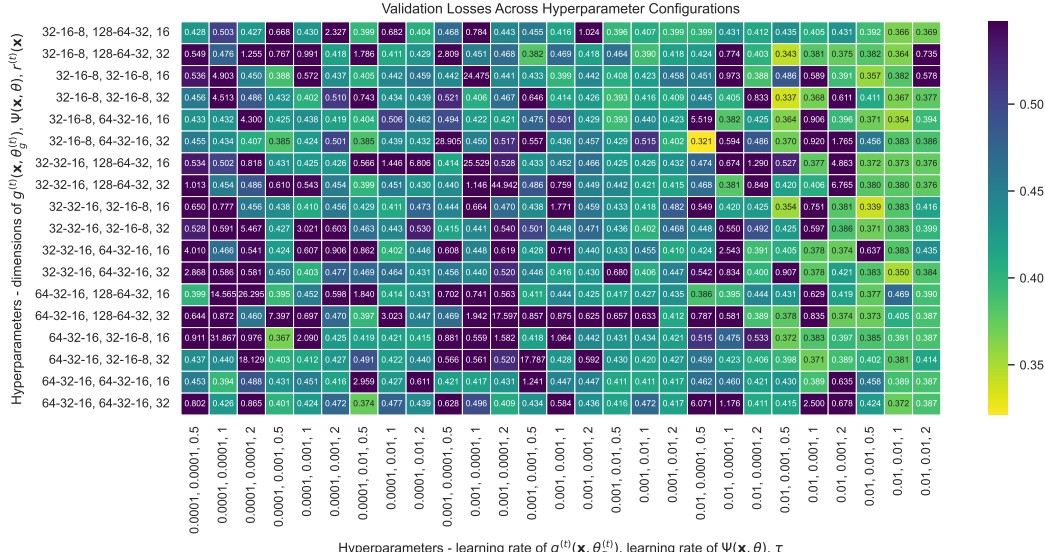

Figure 8: Hyperparameter sensitivity study results of our proposed bi-level knowledge distillation approach on the AKI dataset.

For each ICU stay, we extract 17 physiological variables as input features. Each variable is aggregated across seven predefined time span ranges: the first and last $10\%$, $25\%$, and $50\%$ of the stay duration, as well as the entire time span. Within each time range, we compute six statistical measures per variable: minimum, maximum, mean, standard deviation, skewness, and the number of recorded measurements. This preprocessing yields a total of $714$ features per sample for subsequent analysis.

The task-specific neural oracle $\mathcal{O}_{\mathrm{nn}}$ is implemented as an MLP with three hidden layers of dimensions 512, 256, and 128, respectively. The unified EHR data fidelity predictor $\Psi$ shares the same architecture, also comprising hidden layers of sizes 512, 256, and 128. The representation dimension of $r^{(t)}(\mathbf{x})$ in the attention subnetwork (Equation 7) is set to 128. We use a learning rate of 0.01 for training $\mathcal{O}_{\mathrm{nn}}$ and 0.001 for $\Psi$. The temperature parameter $\tau$ in Equation 11 is fixed at 2.0. Models are trained for a maximum of 1000 epochs with a batch size of 512. Early stopping is employed based on validation performance, with a patience of 50 epochs.

# F  Supplementary Experimental Results

## F.1  Hyperparameter Sensitivity Study on the AKI Dataset

We perform a comprehensive hyperparameter sensitivity analysis on the AKI dataset collected from National University Hospital in Singapore. The tuned hyperparameters and their corresponding search ranges are summarized in Table 4. The number of training epochs is fixed at 1000, with a batch size of 128. Early stopping is applied if the validation performance does not improve for 50 consecutive epochs. The sensitivity results are presented in Figure 8. As illustrated, the optimal validation performance is achieved when the architecture of $g^{(t)}(\mathbf{x}, \theta_g^{(t)})$ is configured with layer

Table 5: Statistical precision of data Shapley value estimates across different Monte Carlo sample sizes for four prediction tasks on the AKI dataset. Standard Error ($SE$) measures estimation uncertainty, while Rolling Variance ($RV$) indicates convergence stability.

(a) Mortality and RRT dependence prediction tasks.

| Sample Size | Mortality Prediction | | RRT Dependence Prediction | |
|---|---|---|---|---|
| | SE ($\times 10^{-5}$) | RV ($\times 10^{-9}$) | SE ($\times 10^{-5}$) | RV ($\times 10^{-8}$) |
| 1,000 | 7.88 | 3.63 | 13.57 | 5.88 |
| 2,000 | 5.56 | 1.75 | 10.60 | 2.85 |
| 5,000 | 3.51 | 0.69 | 7.57 | 1.12 |
| 10,000 | 2.48 | 0.34 | 5.79 | 0.56 |
| 20,000 | 1.76 | 0.17 | 4.23 | 0.28 |
| 50,000 | 1.12 | 0.07 | 2.79 | 0.11 |
| 100,000 | 0.79 | 0.03 | 2.01 | 0.06 |

(b) New or progressive CKD and Stage 5 CKD onset prediction tasks.

| Sample Size | New or Progressive CKD Prediction | | Stage 5 CKD Onset Prediction | |
|---|---|---|---|---|
| | SE ($\times 10^{-5}$) | RV ($\times 10^{-9}$) | SE ($\times 10^{-5}$) | RV ($\times 10^{-8}$) |
| 1,000 | 16.19 | 1.89 | 24.48 | 6.42 |
| 2,000 | 11.80 | 0.91 | 17.78 | 3.09 |
| 5,000 | 7.71 | 0.36 | 11.83 | 1.21 |
| 10,000 | 5.51 | 0.18 | 8.49 | 0.60 |
| 20,000 | 3.93 | 0.09 | 6.04 | 0.30 |
| 50,000 | 2.51 | 0.04 | 3.83 | 0.12 |
| 100,000 | 1.78 | 0.02 | 2.72 | 0.06 |

dimensions [32, 16, 8], and $\Psi(\mathbf{x}, \theta)$ with [64, 32, 16]. The representation dimension of $r^{(t)}(\mathbf{x})$ is set to 32. The learning rates for $g^{(t)}(\mathbf{x}, \theta_g^{(t)})$ and $\Psi(\mathbf{x}, \theta)$ are set to 0.01 and 0.0001, respectively. The temperature parameter $\tau$ is fixed at 0.5. This hyperparameter configuration is subsequently applied to the test dataset for reporting the final evaluation results.

## F.2    Evaluation of $\mathcal{O}_{\text{ds}}$'s Approximation on the AKI Dataset

Although computing exact data Shapley values would, in principle, yield higher accuracy than permutation-based approximations, the exact computation is known to be #P-hard [72, 17], rendering it infeasible for large-scale, real-world datasets such as those used in our experiments. In practice, permutation sampling remains a widely adopted and theoretically grounded approximation strategy for data valuation [26, 65].

To assess the influence of this approximation in $\mathcal{O}_{\text{ds}}$, we conduct 100,000 independent Monte Carlo simulations to estimate data Shapley values and evaluate convergence using two complementary statistical metrics across four post-AKI progression tasks. The results are summarized in Table 5. **Standard Error** ($SE = \sigma/\sqrt{n}$) is used to construct confidence intervals for the estimated values, following $\bar{x} \pm 1.96 \cdot SE$ for the 95% confidence level. The empirical decay of $SE$ at the rate of $O(n^{-1/2})$ is consistent with the Central Limit Theorem, and the final $SE$ falls below one percent across all tasks. **Rolling Variance** ($RV$) measures temporal stability by computing the variance of cumulative means over sliding windows of 100 samples, serving as a practical indicator of when additional samples yield negligible improvements. The observed monotonic decline in $RV$ demonstrates the statistical stabilization of the data Shapley value estimates.

Collectively, the aforementioned analysis provides both theoretical grounding and empirical evidence supporting the reliability of the permutation-based data Shapley value approximation employed by $\mathcal{O}_{\text{ds}}$ in later stages of this study.

Table 6: Comparison between our approach and variant teacher fusion strategies.

| Fusion Strategy | Loss | Strategy Description |
|---|---|---|
| **Our approach** | **0.008135** | Attention-based aggregation as defined in Equation 6 |
| Random weights | 0.010720 | Fixed random weights across teachers |
| Simple average | 0.028935 | Equal-weight aggregation across teachers |
| Top-2 teachers | 0.026531 | Aggregation of the two best-performing teachers |
| Top-1 teacher | 0.144956 | Use of the single best-performing teacher only |

Table 7: Comparison between our approach and variant objective loss functions.

| Configuration | Loss | Modification Details |
|---|---|---|
| **Our approach** | **0.008135** | Full objective loss as defined in Equation 12 |
| No entropy constraint | 0.024595 | $\mathcal{L}_{\text{ent}}$ term removed from objective |
| No similarity constraint | 0.029499 | $\mathcal{L}_{\text{sim}}$ term removed from objective |
| Static weighting | 0.104307 | Fixed $\lambda_{\text{kd}}, \lambda_{\text{ent}}, \lambda_{\text{sim}}$ in Equation 12 |
| High temperature | 0.017665 | Increased temperature for softer distributions ($\tau = 2.0$) |
| Low temperature | 0.109919 | Decreased temperature for sharper distributions ($\tau = 0.1$) |

## F.3 Ablation Study on the AKI Dataset

We first clarify that the knowledge distillation process from $\mathcal{O}_{\text{ds}}$ to $\mathcal{O}_{\text{nn}}$—responsible for deriving application-specific data valuation for downstream analysis—is an integral component of our approach and cannot be removed. We then compare our full approach against several weakened variants of the subsequent distillation step from $\mathcal{O}_{\text{nn}}$ to $\Psi$. As summarized in Table 6, our approach, which integrates an attention-based aggregation strategy, consistently achieves the lowest loss. These results validate the effectiveness of the proposed teacher fusion mechanism and highlight the necessity of employing a principled aggregation strategy during knowledge distillation.

We further perform a detailed ablation analysis to examine the contribution of each component in the overall objective function (Equation 12). In particular, we evaluate the effects of removing individual loss terms, disabling the dynamic weighting mechanism, and varying the temperature hyperparameter. As reported in Table 7, our full approach achieves the best performance, while any component's removal or alteration results in noticeable degradation. These findings collectively demonstrate that the interplay among loss terms, dynamic weighting, and appropriate temperature calibration is critical to achieving optimal knowledge distillation performance.

## F.4 Evaluation of Controlled Deviation Injection on the MIMIC-III Dataset

In this section, we evaluate the effectiveness of the proposed $\Psi$ for detecting deviations in EHR data using the MIMIC-III dataset. We adopt a controlled deviation injection experiment similar to that conducted on the AKI dataset (Section 4.2), introducing deviations with magnitudes ranging from $0.1\sigma$ to $5\sigma$. The comparative results between $\Psi$ and baseline methods are presented in Figure 9.

Consistent with the observations on the AKI dataset (Figure 3), both the baselines and $\Psi$ exhibit improved AUC performance as the deviation magnitude increases, which aligns with the expectation that larger perturbations are more easily detectable. However, $\Psi$ demonstrates superior sensitivity to small deviations. Specifically, it achieves an AUC of $0.91$ when the deviation is as small as $0.1\sigma$, and this performance further improves to $0.94$ at $5\sigma$. In contrast, most baseline methods fail to respond effectively at lower deviation levels and do not reliably distinguish between perturbed and unperturbed samples.

Among the baselines, One-Class SVM, Local Outlier Factor, and $k$-Means Distance show negligible response until the deviation exceeds $3\sigma$, and their performance remains suboptimal even at $5\sigma$. PCA Reconstruction Error outperforms these methods but only achieves an AUC of 0.8 at $5\sigma$. Gaussian Mixture Model is the strongest baseline on MIMIC-III, reaching a competitive AUC under $5\sigma$, yet it remains insensitive to deviations smaller than $1\sigma$, highlighting its limited robustness relative to $\Psi$.

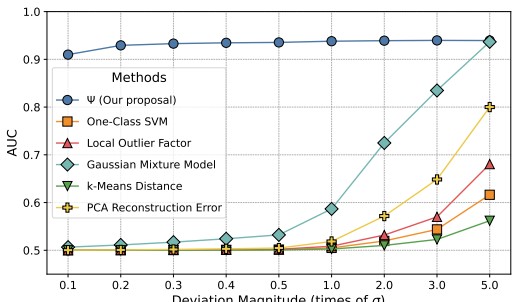
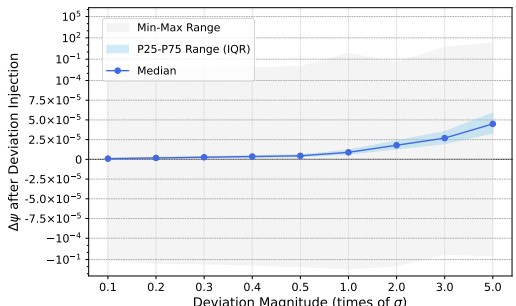

Figure 9: Performance comparison between $\Psi$ and baselines for EHR data deviation detection on the MIMIC-III dataset.

Figure 10: Impact of deviation magnitudes on $\Delta\Psi$ (data fidelity decline) after deviation injection on the MIMIC-III dataset.

Notably, in contrast to the AKI results, the baseline methods on MIMIC-III consistently fall short of $\Psi$'s performance, even at higher deviation magnitudes. This suggests that the MIMIC-III dataset, comprising 25 tasks (phenotypes to classify), poses a more complex deviation detection challenge. Nevertheless, $\Psi$ maintains high performance across the entire range of deviation magnitudes, demonstrating the effectiveness of bi-level knowledge distillation from task-specific data Shapley oracles in enhancing EHR deviation detection.

### F.5 Evaluation of $\Psi$'s Output Sensitivity on the MIMIC-III Dataset

We further assess the output sensitivity of $\Psi$ on the MIMIC-III dataset by measuring the fidelity decline, denoted as $\Delta\Psi$, under varying magnitudes of injected deviations from $0.1\sigma$ to $5\sigma$. The results are summarized in Figure 10, which reports the median, interquartile range (IQR), and minimum-maximum range of $\Delta\Psi$.

Consistent with the observations on the AKI dataset (Figure 4), $\Delta\Psi$ increases monotonically as the deviation magnitude grows. Notably, even when $\Delta\Psi$ is close to zero, $\Psi$ remains effective at detecting subtle deviations by leveraging the sign of $\Delta\Psi$ as a reliable indicator. This is corroborated by the high AUC observed in Figure 9 at small deviation levels, reinforcing the ability of $\Psi$ to identify early-stage deviations that are typically overlooked by baseline methods. This property is particularly valuable for early warning applications in EHR data analytics.

Additionally, we observe that the absolute values of $\Delta\Psi$ on the MIMIC-III dataset are generally smaller than those on the AKI dataset, suggesting that the output sensitivity of $\Psi$ varies less markedly with increasing deviation magnitude in this setting. Furthermore, $\Delta\Psi$ can be negative under small deviations, indicating occasional prediction errors by $\Psi$ in extreme cases. These findings underscore the increased difficulty of the MIMIC-III scenario, consistent with the baseline performance degradation reported in Appendix F.4. Nevertheless, $\Psi$ continues to deliver robust deviation detection performance, even in this more complex and diverse application context.

### F.6 Comparison with Rule-based Methods on the MIMIC-III Dataset

Existing rule-based methods that incorporate domain knowledge—particularly for laboratory test data—mainly focus on detecting values that fall outside normal physiological ranges [71, 85, 83]. In this study, we refer to a publicly available list of clinically validated variable ranges associated with the MIMIC-III multi-task learning benchmark. This list, developed in consultation with clinical experts, reflects their domain understanding of physiologically plausible measurement intervals [83]. Each variable is defined by upper and lower bounds specifying the physiologically acceptable range, and any observed value that lies outside these thresholds can be flagged as abnormal. This constitutes a straightforward, domain-informed rule-based detection method.

We summarize key features from our dataset and report their empirical means and standard deviations (std $\sigma$) alongside the corresponding clinically valid ranges in Table 8. It is noteworthy that the clinically defined ranges typically span more than $10\sigma$, which is substantially broader than the deviations considered in our experiments (limited to perturbations up to $5\sigma$). Consequently, such rule-based methods can only identify overt abnormalities and are insufficient for detecting more

Table 8: Statistics and clinically valid value ranges for key features on the MIMIC-III dataset.

| Feature | Mean | Std | Valid Low | Valid High | Valid Range/Std |
|---|---|---|---|---|---|
| Diastolic blood pressure | 59.57 | 15.26 | 0.00 | 375.00 | 24.57 |
| Fraction inspired oxygen | 0.47 | 0.26 | 0.21 | 1.00 | 3.08 |
| Glascow coma scale total | 11.44 | 3.74 | 3.00 | 15.00 | 3.21 |
| Glucose | 138.27 | 67.38 | 33.00 | 2000.00 | 29.19 |
| Heart rate | 101.39 | 32.85 | 0.00 | 350.00 | 10.66 |
| Mean blood pressure | 78.37 | 18.20 | 14.00 | 330.00 | 17.36 |
| Oxygen saturation | 96.82 | 4.25 | 0.00 | 100.00 | 23.55 |
| Respiratory rate | 24.65 | 14.68 | 0.00 | 300.00 | 20.44 |
| Systolic blood pressure | 119.65 | 25.74 | 0.00 | 375.00 | 14.57 |
| Temperature | 36.90 | 0.85 | 26.00 | 45.00 | 22.43 |

subtle yet clinically meaningful deviations. In contrast, our proposed $\Psi$ is specifically designed to detect small-magnitude deviations often overlooked yet potentially impactful (see Figures 3 and 9). This capability enables earlier clinical intervention and supports more reliable data quality assurance in real-world healthcare settings.

# G    Broader Impact

The proposed bi-level knowledge distillation approach enables reliable detection of potential data deviations in EHR data arising from sources such as pre-analytical variability, documentation errors, or unvalidated data sources. By assessing the EHR data fidelity, the approach enhances the reliability and accuracy of downstream clinical decisions and interventions. As such, it represents a promising direction for improving data acquisition, collection, and recording protocols and may serve as a foundation for future error correction and calibration mechanisms.

Beyond these technical contributions, it is essential to involve clinicians and healthcare professionals when applying EHR deviation detection in clinical practice. In particular, a decline in data fidelity may not solely stem from artifacts or errors—it may also reflect meaningful underlying physiological dynamics related to iatrogenic reasons, or medical interventions, reflecting the complex interplay of various acute medical conditions that occur concurrently in real-world patients [78]. In such cases, additional contextual information or multimodal data sources may be required to interpret these deviations accurately and to understand the patient's clinical condition. For example, from a nephrological perspective, acute dialysis introduces substantial fluctuations in renal function measurements. These fluctuations do not align with the progressive trajectory of worsening or severe renal failure but instead reflect the treatment-induced modulation of physiological parameters. Consequently, our approach for detecting EHR data deviations may not be limited to removing erroneous entries or correcting recording mistakes. Rather, it can facilitate disease-specific analysis by filtering out concurrent medical noise that arises from complex patient care processes, where multiple acute diseases, transfusions, infusions, and medications jointly influence observed trends. Addressing these challenges necessitates collaborative efforts between computational researchers and domain experts, and highlights an important open area for future investigation.

