# OpenReview forum: "Detecting Data Deviations in Electronic Health Records"
_NeurIPS.cc/2025/Conference — NeurIPS 2025 poster_

### Official Review · Reviewer_8yxT · 2025-07-02

**Clarity:** 2
**Significance:** 2
**Originality:** 4
**Rating:** 4
**Confidence:** 3

**Summary:**

This paper addresses the problem of detecting EHR discrepancies between recorded values and the patient’s true physiological state (ie "data deviation") in Electronic Health Records (EHRs). They propose a multi-step framework to estimate whether the data is 'good' based on first training task-specific models (using Shapley values) to measure how much each data point contributes to downstream clinical tasks and then then combining those models into a single, general-purpose predictor. This final model assigns a "fidelity score" to new data points. If that score drops sharply after a small change to the input, it signals that the data may be unreliable. The method is tested on both private and public (in the appendix) datasets.

**Questions:**

- None of the models are directly looking at the time series, though such information would probably be very valuable for the underlying question of whether the collected data is a valid reflection of the patient’s state. Was this considered? If so, why was it not leveraged?
- Intuitively why were the elements of this model combined? For instance, why did you think to use Shapley values as a first step?
- Am I correct in my understanding that this approach does not verify whether the EHR data is an accurate measurement of patient physiology but rather it measures whether the data appears useful in downstream models? These are not always the same. Misleading data can sometimes improve task predictions if it correlates spuriously with outcomes (e.g., due to confounding).

**Ethical Concerns:**

["NO or VERY MINOR ethics concerns only"]

**Final Justification:**

I thank the authors for their clarifications and responses.

I had initially misunderstood the controlled deviation injection experiment. On re-reading, I see that it is indeed present, though I would suggest slightly clearer framing in the main text to aid comprehension.

I'm still unconvinced about both:
W1) how generalizable are these data corruptions?
Q3) is this method detecting data fidelity or feature impact on performance.

For Q1, in my experience, perturbed numeric values is one kind of data entry, though not the most egregious. There are non-smooth data entry errors such as using the wrong units or entering an extra digit (eg someone who ways 2122 pounds). That said, this method doesnt need to account for *all* data errors. I merely speculate that the utility of this approach may be more narrow than the authors believe.

On Q3, I think the authors and I agree on the mechanics: the approach identifies data elements whose perturbation most affects downstream model utility. Where we differ is in interpretation. The authors argue that their goal is not to enhance downstream performance, and thus their approach measures fidelity rather than utility. My view is that (even with a multi-task setup) model-derived utility remains an indirect proxy for fidelity. The method does not verify whether the EHR data reflect accurate physiological measurements (which is understandably infeasible), but instead assesses how useful the data appears to be for model performance. Whether that is an acceptable stand-in for "fidelity" remains open to interpretation.

I will maintain my score of borderline accept.

**Limitations:**

yes

**Quality:**

3

**Strengths And Weaknesses:**

Strengths
- This area (identifying and potentially addressing incorrect input values) is very valuable for every domain, especially including healthcare which has many opportunities for noise to occur.
- Section 4.4's case studies did a very good job explaining how one could identify data quality issues (eg two labs that should probably move together actually end up diverging in their series).
- I appreciated seeing baselines to compare against to illustrate the task & alternative solutions.


Weaknesses
- This area, while important, is hard to measure whether particular contributions are meaningful. For instance, the corruptions to the data you chose might not be meaningful or generalizable to other data collection issues.
- Some of the baselines were weak. Many are designed to identify global outliers, rather than the e.g. multi-lab time series correlations illustrated in the case study. I would’ve liked to see something like “real vs corrupted” classifier trained to predict actual data points (label=0) vs the corrupted data points you create (label=0).

Minor Weaknesses
- The writing in the paper often used technical jargon (which made it hard to follow on first pass) when simpler explanations could've worked just as well if not better. As an illustration, instead of "Knowledge is distilled from the data Shapley oracle  𝑂 𝑑 𝑠 O  ds ​  … into a corresponding neural oracle  𝑂 𝑛 𝑛 O  nn ​  … trained jointly in a multi-task learning setting." I would have benefitted from seeing early on "We train a set of models to mimic the Shapley values efficiently, one for each prediction task."
- Until Section 4.4 I was a little confused about the inputs/outputs of the task setup (eg is it predicting the fidelity of a single data element vs a batch of results vs something else). This could have been made more clear earlier on.

---

> ### Author Rebuttal · Authors · 2025-07-31
>
> **W1.** We respectfully disagree with the concern regarding the relevance and generalizability of our perturbation strategy. In clinical practice, point-of-entry validation for EHR data is essential due to the risk of serious downstream consequences from erroneous inputs. However, existing validation practices rely largely on coarse plausibility checks—such as broad value ranges or thresholds based on standard deviations—which are insufficient for detecting subtle yet clinically significant deviations (see also our response to **Reviewer opND's comment W1**).
>
> Our strategy, which perturbs feature values by their standard deviation $\sigma$, aligns with established clinical quality control practices and offers a clinically interpretable way to model abnormal laboratory readings [c, d, e]. Rather than an arbitrary choice of corruptions, what our work contributes is a principled, sensitive framework that addresses the limitations of current checks by enabling pre-hoc detection and alerting of meaningful deviations prior to downstream contamination.
>
> This design reflects realistic data fidelity issues observed in operational EHR workflows. Our clinicians have confirmed that our experimental setup is not only meaningful but also directly aligned with the operational needs and safety requirements of healthcare data entry. We will revise the paper to articulate this clinical grounding and to discuss the broader applicability of our approach beyond the specific perturbations evaluated.
>
> **W2.** We are not entirely certain if we have interpreted this comment correctly and would appreciate any clarification if our current understanding is inaccurate.
>
> We would like to first clarify a potential misunderstanding on the experimental evaluation. Our paper does include the suggested “real vs corrupted” classification experiment to distinguish actual data samples from perturbed ones. In our controlled deviation injection setup, we (i) perturb data with the feature's standard deviation $\sigma$, (ii) generate paired samples—original (label = 0) and perturbed (label = 1), and (iii) construct a labeled benchmark dataset for fidelity evaluation. We then compute the expected decline in EHR data fidelity $\Delta \Psi$ and use its sign to indicate whether a deviation is detected.
>
> As shown in Figure 3, our approach outperforms anomaly detection baselines, particularly in detecting small-magnitude deviations. We have also compared against domain-knowledge-based rules and presented a detailed ablation study assessing different constituent components. Please refer to our response to **Reviewer opND’s comment W1** for related discussion. We remain open to additional suggestions on methods relevant to our setting and will incorporate such comparisons where feasible.
>
> **W3.** We will revise the paper to introduce our core idea using clearer and more accessible descriptions before presenting the detailed design and terminology. We believe this restructuring will improve clarity and make our contributions more approachable to a broader audience.
>
> **W4.** Our approach outputs a predicted EHR data fidelity score $\Psi$ for each data sample. To detect deviations, we compute the expected EHR data fidelity decline $\Delta \Psi$ and use its sign as an indicator.
>
> In the controlled deviation injection experiment (Figure 3), we simulate physiologically plausible deviations by perturbing features with multiples of their standard deviation $\sigma$, forming paired samples ⟨original (label = 0), perturbed (label = 1)⟩. We apply our approach to compute $\Delta \Psi$ for each pair and use the sign to identify deviations. Results demonstrate the strong detection performance of our approach compared to baselines.
>
> In the output sensitivity experiment (Figure 4), we use the same paired samples as input with varying magnitudes of the injected deviations. The resulting $\Delta \Psi$ values demonstrate that our approach reliably captures subtle deviations, underscoring its practical utility in early-stage deviation detection for clinical applications.
>
> We acknowledge that the rationale and structure of these experiments may not be immediately clear on a first reading. To address this, we will revise the paper to introduce the experimental setups and key design choices earlier, improving clarity and readability.
>
> **Q1.** We follow the established MIMIC-III benchmark for data processing, aggregating 17 physiological variables across seven predefined time ranges: the first and last 10%, 25%, 50% of the ICU stay, and the entire duration. For each variable within each range, six statistics are computed—minimum, maximum, mean, standard deviation, skewness, and count—yielding 714 features per sample. This preprocessing strategy has demonstrated strong performance on benchmark MIMIC-III tasks in prior work and is adopted here to ensure consistency. As our focus is on pre-hoc detection of data deviations at the point of entry rather than optimizing preprocessing, we do not explore alternative preprocessing strategies.
>
> From a practical standpoint, incorporating more advanced time series models would exacerbate the already significant computational cost of estimating data Shapley values. Exact data Shapley computation is intractable for large-scale EHR data due to its exponential complexity [14]. Even with Monte Carlo approximation, the process requires repeated model retraining for each sample across multiple permutations. To address this, we adopt an amortized modeling approach, training a neural network to approximate per-sample data Shapley values [1, 6]. While this strategy reduces inference-time cost, it still requires computing ground-truth data Shapley values during training. Given these constraints, more complex time series models are currently infeasible. Improving the efficiency of Shapley-based data valuation continues to be an important and active area of research [46].
>
> **Q2.** Our methodology is designed in a top-down, stepwise manner, guided by the central goal of detecting EHR data deviations. Below, we clarify the rationale behind each component.
>
> To begin, detecting deviations requires a principled measure of data fidelity. Data valuation methods—particularly data Shapley values—are increasingly used to quantify sample-level contributions to learning models [14, 25, 46, 54, 57, a, b], making them a natural proxy for data fidelity. However, data Shapley values are inherently task-specific and thus insufficient for capturing a more comprehensive notion of fidelity across a patient’s EHR data.
>
> To overcome this, we adopt a multi-task perspective that distills task-specific data Shapley values into a unified EHR data fidelity predictor. This integration enables a more comprehensive assessment by incorporating insights across multiple clinical tasks.
>
> In summary, we propose two levels of knowledge distillation:
> (i) from computationally intensive data Shapley values to task-specific neural oracles, to address efficiency concerns (see Sections 3.1 and 3.2), and
> (ii) from the task-specific neural oracles to a unified EHR data fidelity predictor that captures cross-task insights.
>
> The motivation for this design and the necessity of each knowledge distillation level are detailed in Section 2. We have also included an ablation study evaluating the impact of different components (see our response to **Reviewer opND’s comment W1**), and a sanity check validating the approximation of data Shapley values (see our response to **Reviewer oTHf's comment W1**).
>
> We will incorporate the above discussion into the paper to further clarify the rationale behind our proposed methodology and facilitate readers' understanding of our technical design choices.
>
> **Q3.** We would like to clarify that our objective is fundamentally different from enhancing the analytic performance of downstream models. Instead, our focus lies in **pre-hoc detection of potential data deviations at the point of entry**, with the explicit goal of preventing such records from contaminating the EHR system. Our approach raises alerts upon detecting deviations, enabling clinicians and healthcare staff to take timely actions, such as investigating root causes or refining data entry workflows, thereby ensuring overall data fidelity.
>
> We understand that this comment likely arises from our use of data Shapley values as a basis for constructing a fidelity measure. Data Shapley values estimate a sample’s contribution to downstream tasks, which does not always equate to its alignment with true physiological measurements. This distinction is well noted, and addressing this gap forms the core innovation of our work. Specifically, we integrate task-specific data Shapley values derived from multiple clinically relevant tasks into a unified fidelity predictor, which captures the holistic quality of EHR data beyond any single downstream objective. By combining these multi-task contributions, our approach reduces the risk of over-reliance on task-specific correlations and produces a more robust indicator of data fidelity. Experimental results and expert evaluations from clinicians confirm that this integrated approach effectively identifies deviations and highlights clinically meaningful data quality issues observed in real-world cases.
>
> We also acknowledge the complex relationship between misleading data and downstream performance. Some deviations degrade performance by introducing noise or systematic errors, while others may inadvertently boost performance via spurious correlations or confounding factors. Although exploring such task-dependent effects is beyond the primary scope of this study, we agree that this is an important perspective and will revise the paper to (i) restate our emphasis on data fidelity rather than downstream performance enhancement, and (ii) expand the discussion to reflect the broader implications and impact of misleading data on downstream performance.

---

### Official Review · Reviewer_opND · 2025-07-03

**Clarity:** 4
**Significance:** 3
**Originality:** 3
**Rating:** 5
**Confidence:** 4

**Summary:**

The authors study the novel problem of detecting data deviation in Electronic Health Records, which has implications on the reliability of patient data for downstream tasks. They propose a 2-step knowledge distillation approach that calculator Shapley values for individual tasks, and eventually learns a general neural model for data valuation. They use controlled deviation injection to show the ability for their method to identify anomalies on data from both public and private EHR records.

**Questions:**

One might ask if the goal is to train a good model on EHR data, then can we just assume that data deviation is random noise that a good model should ignore? Are there any experiments you could run outside of controlled injection studies to show that this actually leads to a performance difference?

**Ethical Concerns:**

["NO or VERY MINOR ethics concerns only"]

**Final Justification:**

Thank you for the thorough response, and I appreciate the clarifications surrounding pre-hoc detection. While the authors mention that clinicians have confirmed the practice necessity of pre-hoc methods, it would be great if the paper reflected this as the introduction still largely motivates the work with the potential of EHR analytics.

**Limitations:**

Yes

**Quality:**

4

**Strengths And Weaknesses:**

The authors study a very interesting and novel problem, and the paper is very well-written. The overall approach considering Shapley scores makes intuitive sense, and the experiments are thorough across several interesting tasks.

Some weaknesses include:

W1. Lack of studying ablations of the proposed method (e.g. removing KD steps) versus just comparing against other baselines. Furthermore, are there any simple rule-based approaches leveraging the domain knowledge that could be used as a baseline?

W2. Clarification whether the primary goal is to remove data for the purpose of downstream prediction, or other uses. Since the method only outputs an anomaly score,  it doesn’t help shed light on why any deviation occurred. The case studies in the paper were really useful, but primarily seemed to highlight an issue with duplication: meanwhile, the paper’s introduction highlights a wider source of data deviation. Could the authors provide more insights from the data on the types of deviation error?

W3. Discuss any limitations around removing data deviation points – similar to arguments around how incorrect data in LMs are actually useful for calibration and misinformation detection (or training on toxic text helps LMs know toxic vs. non-toxic and generate less toxic text), what if a model built on EHR data might actually benefit from being aware of potential data deviations? Are there concerns around robustness?

---

> ### Author Rebuttal · Authors · 2025-07-31
>
> **W1.** **Ablation study results.** We first clarify that the knowledge distillation process from $\mathcal{O} _ {ds}$ to $\mathcal{O} _ {nn}$—which enables the derivation of application-specific data valuation for downstream analysis—is an essential component of our approach and cannot be removed. In response to the suggested ablation study, we have compared our proposal against several weakened variants of the subsequent distillation step from $\mathcal{O} _ {nn}$ to $\Psi$. As shown in the table below, our approach, which incorporates an attention-based aggregation strategy, consistently achieves the lowest loss. These results underscore the effectiveness of the proposed teacher fusion mechanism and demonstrate the necessity of a principled aggregation strategy in this knowledge distillation process.
>
> |Variant|Loss|Strategy Description|
> |-|-|-|
> |**Our approach**|**0.008135**|Attention-based aggregation as defined in Equation (6)|
> |Random weights|0.010720|Fixed random weights across teachers|
> |Simple average |0.028935|Equal-weight aggregation across teachers|
> |Top-2 teachers|0.026531|Aggregation of the two best-performing teachers|
> |Top-1 teacher|0.144956|Use of the single best-performing teacher only|
>
> We have further conducted a fine-grained ablation analysis to assess the contribution of each component in the overall objective loss in Equation (12). Specifically, we examine the effects of removing individual loss terms, disabling the dynamic weight adjustment mechanism, and modifying the temperature setting. As shown in the table below, our full approach achieves the best performance. Performance degrades noticeably when any component is removed or altered, confirming the importance of each design choice. These results collectively underscore that the synergy among loss terms, dynamic weighting, and appropriate temperature calibration is critical for optimal knowledge distillation.
>
> |Variant|Loss|Strategy Description|
> |-|-|-|
> |**Our approach**|**0.008135**|Full objective loss as defined in Equation (12)|
> |Without entropy constraint|0.024595|$\mathcal{L}_{ent}$ removed from objective|
> |Without similarity constraint|0.029499|$\mathcal{L}_{sim}$ removed from objective|
> |Fixed weights|0.104307|Static $\lambda_{kd}$, $\lambda_{ent}$, $\lambda_{sim}$ in Equation (12)|
> |High temperature|0.017665|Temperature set to $\tau=2.0$|
> |Low temperature|0.109919|Temperature set to $\tau=0.1$|
>
> **Rule-based approaches leveraging domain knowledge.** Existing rule-based methods that leverage domain knowledge—particularly for laboratory test results—primarily focus on detecting "values outside normal physiological ranges" [c, d, e]. For this study, we have identified a publicly available list of clinically validated variable ranges associated with the MIMIC-III multi-task learning benchmark. This list was developed in consultation with clinical experts and reflects their domain knowledge of physiologically plausible measurement ranges [e]. Each variable in this list is associated with predefined upper and lower bounds that specify a physiologically acceptable range. Observed values falling outside these thresholds can be flagged as abnormal, thus forming a simple domain-informed rule-based detection method.
>
> We have summarized key features from our dataset and reported their empirical means and standard deviations ($\sigma$) alongside the corresponding valid ranges in the table below. Notably, the clinically defined ranges typically exceed $10\sigma$, far beyond the scale of deviations considered in our experiments (which are limited to perturbations up to $5\sigma$). As a result, such rule-based methods are only capable of identifying overt anomalies and are inadequate for capturing more subtle yet meaningful deviations. In contrast, our proposed $\Psi$ is specifically designed to detect small-magnitude deviations often overlooked yet potentially impactful (see Figure 3). This capability supports timely clinical intervention and contributes to more robust data quality assurance in real-world healthcare settings.
>
> |Feature|Mean|Std|Valid Low|Valid High|Valid Range/Std|
> |-|-|-|-|-|-|
> |Diastolic blood pressure|59.57|15.26|0.00|375.00|24.57$\sigma$|
> |Fraction inspired oxygen|0.47|0.26|0.21|1.00|3.08$\sigma$|
> |Glascow coma scale total|11.44|3.74|3.00|15.00|3.21$\sigma$|
> |Glucose|138.27|67.38|33.00|2000.00|29.19$\sigma$|
> |Heart Rate|101.39|32.85|0.00|350.00|10.66$\sigma$|
> |Mean blood pressure|78.37|18.20|14.00|330.00|17.36$\sigma$|
> |Oxygen saturation|96.82|4.25|0.00|100.00|23.55$\sigma$|
> |Respiratory rate|24.65|14.68|0.00|300.00|20.44$\sigma$|
> |Systolic blood pressure|119.65|25.74|0.00|375.00|14.57$\sigma$|
> |Temperature|36.90|0.85|26.00|45.00|22.43$\sigma$|
>
> > [c] Shi, Xi, et al. "An automated data cleaning method for Electronic Health Records by incorporating clinical knowledge." BMC Medical Informatics and Decision Making 21.1 (2021): 267.
>
> > [d] Wang, Zhan, et al. "Rule-based data quality assessment and monitoring system in healthcare facilities." Improving Usability, Safety and Patient Outcomes with Health Information Technology. IOS Press, 2019. 460-467.
>
> > [e] Wang, Shirly, et al. "Mimic-extract: A data extraction, preprocessing, and representation pipeline for mimic-iii." Proceedings of the ACM conference on health, inference, and learning. 2020.
>
> **W2.** We clarify that the primary goal of our proposal is to enable **pre-hoc detection of potential data deviations at the point of data entry** into the EHR system during clinical practice. The aim is not data cleaning or post-hoc removal for downstream analytics, but rather early identification of anomalies at the time of data recording. By highlighting entries with possible deviations as they are introduced, our approach supports real-time quality control and reduces the likelihood of erroneous data entering the system. This leads to improved EHR data quality, which may in turn enhance the performance of downstream analytical tasks. However, this improvement is a secondary benefit; our central goal remains the reliable assessment of data fidelity in situ, to prevent flawed data from influencing clinical decision-making for real-world patients.
>
> We also clarify that the presented case studies go beyond simple duplication scenarios. For instance, the Point-of-Care Testing (POCT) laboratory tests featured in our case studies are conducted at or near the site of patient care rather than in centralized laboratories, with the aim of enabling rapid clinical decisions. When multiple test results for the same feature are recorded within a short timeframe and exhibit inconsistencies or abnormalities, the underlying causes can be multifactorial. These include misattribution of results to the wrong patient, errors in sample collection, device malfunctions in uncontrolled environments, or testing conducted at inappropriate times relative to clinical interventions or medication schedules.
>
> Our clinicians confirm the practical necessity and clinical value of detecting such potential data deviations in a pre-hoc manner. It is emphasized that alerting healthcare workers to possible issues at the time of testing can help prevent erroneous data from being integrated into subsequent clinical workflows, thereby improving the robustness of patient care and decision-making.
>
> **W3 & Questions.** We would like to clarify that our primary goal is neither post-hoc data removal or cleaning, nor the development of high-performing EHR analytic models. Instead, our focus is on **pre-hoc detection of potential data deviations prior to their integration into the EHR system** in clinical practice. Such early detection enables clinicians and healthcare workers to intervene promptly, facilitating root cause investigation and the refinement of clinical workflows—for example, by improving device stability or updating standard operating procedures.
>
> While we acknowledge the possibility of leveraging our deviation detection outputs as auxiliary information in EHR analytics—for example, using them as pseudo-labels to inform model training—we note that the impact of such inherent noise or deviations is nuanced. Depending on the context, these signals could either degrade performance or promote model robustness. Exploring this trade-off, however, goes beyond the primary scope of our current study. We will revise the paper to clarify these points and incorporate further discussion accordingly.

---

> > ### Comment · Reviewer_opND · 2025-08-05
> >
> > Hi, Thank you for the thorough response, and I appreciate the clarifications surrounding pre-hoc detection. While the authors mention that clinicians have confirmed the practice necessity of pre-hoc methods, it would be great if the paper reflected this as the introduction still largely motivates the work with the potential of EHR analytics. I still believe it is an interesting paper for NeurIPS,  and am maintaining my score.

---

> > > ### Author Response · Authors · 2025-08-05
> > >
> > > We appreciate your constructive reviews and recognition of our work, which are valuable for further improving the paper. As suggested, we will revise the paper to better highlight the necessity of pre-hoc methods and incorporate the discussions during the rebuttal.

---

### Official Review · Reviewer_oTHf · 2025-07-04

**Clarity:** 3
**Significance:** 2
**Originality:** 2
**Rating:** 3
**Confidence:** 4

**Summary:**

The paper addresses the under-explored challenge of data deviation, discrepancies between recorded EHR entries and a patient’s true physiological state, which diminishes data fidelity and jeopardises downstream clinical analytics. It models fidelity as a task-agnostic property and introduces a bi-level knowledge-distillation pipeline: task-specific data-Shapley values (the “Data Shapley Oracle”) are first approximated by neural oracles and then fused into a unified predictor Ψ for patient-level fidelity estimation. Monte-Carlo Shapley sampling combined with neural amortisation slashes the exponential cost of exact valuation, making near-real-time fidelity scoring feasible on large healthcare datasets. An anomaly is flagged whenever Ψ registers a fidelity drop beyond a learned threshold, enabling alerts for documentation errors, specimen-handling issues, or unreliable wearable-device readings.

**Questions:**

Please refer to the weakness.

**Ethical Concerns:**

["NO or VERY MINOR ethics concerns only"]

**Final Justification:**

My concerns are still remaining. I personally believe this would be a good paper in the future, but it would need further quality improvement. Thus, I would keep my score.

**Limitations:**

Please refer to the weakness.

**Paper Formatting Concerns:**

I don't have any concerns.

**Quality:**

3

**Strengths And Weaknesses:**

Strengths

- The paper tackles the important problem of detecting anomalies in EHR data.
- The manuscript is clearly written and generally easy to follow.

Weaknesses

- The “Data Shapley Oracle” is produced via permutation sampling rather than by computing exact Shapley values. This approximation can introduce non-trivial error that may be further amplified when distilled into the proposed Neural Oracle. The paper should report sanity checks that quantify this gap before treating the oracle as ground truth.
- The amortized Shapley framework may fail under domain shift. Out-of-distribution samples, common in healthcare, where each clinical sub-domain is highly specialized, could yield unreliable value estimates and undermine the proposed approach.
- Definition 1 specifies a deviation threshold, yet in practice, this threshold behaves like a tunable hyperparameter. In real clinical settings, such cut-offs are hard to set, raising questions about whether the fidelity metric can robustly flag true abnormalities in EHR data.
- The data fidelity used in this paper for detecting data deviation is highly dependent on the provided value function. However, the paper lacks explanation and analysis of the sanity of the value function. If the value function losts its sanity in providing prediction, then the entire scoring system will be not realiable and meaningless. The paper is recommended to include either empirical assessment or domain experts evaluation for sanity check.

---

> ### Author Rebuttal · Authors · 2025-07-31
>
> **W1.** While we acknowledge that computing exact data Shapley values would yield higher accuracy than permutation-based sampling, the exact computation incurs exponential complexity and is intractable for the large-scale, real-world datasets used in our experiments. Further, permutation sampling remains a widely adopted and theoretically grounded approximation strategy in this line of research [14, 46].
>
> To evaluate the potential impact of this approximation, we have performed 100K independent Monte Carlo simulations to estimate data Shapley values, employing two complementary metrics to assess convergence with statistical rigor across four post-AKI progression tasks. **Standard Error** ($SE = \sigma/\sqrt{n}$) underpins the construction of confidence intervals: $\bar{x} \pm 1.96 \cdot SE$ provides 95% confidence bounds for the data Shapley values. The observed $O(n^{-1/2})$ decay rate aligns with the Central Limit Theorem, and the final SE reaches sub-percentage levels across all tasks. **Rolling Variance** ($RV$) quantifies temporal stability by measuring the variance of cumulative means over sliding windows of 100 samples. This serves as a practical indicator to assess when additional sampling yields diminishing returns. The observed monotonic decline in $RV$ indicates statistical stabilization of the estimates.
>
> These analyses provide both theoretical justification and empirical support for the reliability of the permutation-based data Shapley value estimates used in our subsequent analyses.
>
> |Sample Count|Mortality Prediction||RRT Dependence Prediction||CKD Prediction||ESKD Onset Prediction||
> |-|-|-|-|-|-|-|-|-|
> || $SE (10^{-5})$ | $RV (10^{-9})$ | $SE (10^{-5})$ | $RV (10^{-8})$ | $SE (10^{-5})$ | $RV (10^{-9})$ | $SE (10^{-5})$ | $RV (10^{-8})$ |
> |1k|7.88|3.63|13.57|5.88|16.19|1.89|24.48|6.42|
> |2K|5.56|1.75|10.60|2.85|11.80|0.91|17.78|3.09|
> |5K|3.51|0.69|7.57|1.12|7.71|0.36|11.83|1.21|
> |10K|2.48|0.34|5.79|0.56|5.51|0.18|8.49|0.60|
> |20K|1.76|0.17|4.23|0.28|3.93|0.09|6.04|0.30|
> |50K|1.12|0.07|2.79|0.11|2.51|0.04|3.83|0.12|
> |100K|0.79|0.03|2.01|0.06|1.78|0.02|2.72|0.06|
>
> **W2.** We thank the reviewer for highlighting the potential influence of domain shift on our amortized Shapley framework. We agree that domain shift—characterized by changes in the underlying data distribution—is a critical and challenging problem, particularly in healthcare settings where new diseases (e.g., COVID-19) or highly specialized sub-domains can result in out-of-distribution (OOD) samples. We acknowledge this as a potential limitation of our study, but emphasize that it is unlikely to affect our current setting for the following reasons.
>
> First, our work focuses on EHR data within a single hospital. Although federated EHR scenarios involving multiple hospitals are valuable, they are beyond the scope of our current study, as we do not manage inter-hospital domain shifts.
>
> Second, within our hospital, data heterogeneity across different departments and diseases is already addressed through our multi-task modeling framework. This approach effectively unifies diverse clinical contexts by incorporating task-specific information without requiring explicit domain adaptation.
>
> Third, while domain shift research often investigates models trained on data from one domain and deployed in another, such a setting is not highly relevant to our routine clinical workflows. Except for emerging infectious diseases or other rare events, our clinical decision support system is typically developed and applied within the same institutional and departmental context.
>
> Finally, our approach is primarily designed for **pre-hoc detection of data deviations at the point of data entry**, aiming to alert clinicians and healthcare staff before erroneous data affects downstream processes. From this perspective, the risk posed by domain shift is minimal because the features of interest—such as frequently recorded vital signs and laboratory results—are consistent across common clinical tasks. Domain shift in rarely used or highly specialized tests (e.g., biomarkers for rare diseases) has far less impact on data quality assurance, which is our main focus.
>
> While we recognize domain shift as an important area for future extension, we believe it does not undermine the contributions or validity of our current work. We will revise the paper to explicitly discuss this potential limitation and outline possible directions for integrating domain adaptation techniques in broader, cross-institutional settings.
>
> **W3.** Yes, the threshold $\delta$ in Definition 1 can be adapted to suit specific practical requirements and deployment scenarios. For instance, in our experiment evaluating the sensitivity of the proposed approach $\Psi$ to varying deviation magnitudes (results shown in Figure 4), we set $\delta = 0$ and interpret the sign of $\Delta \Psi$ as an indicator of whether a deviation is detected. As shown in Figure 4, when the detection objective emphasizes reliable performance within the interquartile range (P25–P75), this zero-threshold setting is sufficient. In particular, the sign of $\Delta \Psi$ consistently provides a stable and effective signal for identifying deviations, even for subtle perturbations near the decision boundary.
>
> We acknowledge that in real-world applications, the threshold $\delta$ should be customized based on domain-specific considerations. We work closely with our frontline clinicians to calibrate this threshold for practical use, balancing detection accuracy with the need to minimize alert fatigue in clinical workflows.
>
> **W4.** To assess the fidelity of EHR data, we leverage data valuation techniques to quantify the contribution of individual data samples. Among various approaches, the data Shapley value has emerged as a prominent method in the data valuation literature [14, 25, 46], and is increasingly recognized as a standard tool for assessing data quality in high-stakes domains such as healthcare [54, 57, a, b]. Accordingly, we adopt the data Shapley value as our data valuation method to derive EHR data fidelity. While the selection of data valuation methods is not the main focus of our work, our proposed approach is readily applicable to other, potentially more advanced, alternatives that may further enhance detection performance.
>
> Given the high computational cost of exact data Shapley value calculation, we employ Monte Carlo sampling for approximation. The validity of this approximation has been discussed in detail in our response to **your comment W1**. Empirical results demonstrate that our approach achieves strong performance in detecting EHR data deviations and provides clinically meaningful insights into potential data quality issues of real-world cases, as validated by our clinicians.
>
> Importantly, our approach is designed as a pre-hoc data fidelity assessment procedure that addresses critical gaps in current EHR workflows by (i) identifying and excluding erroneous records from clinical workflows, thereby mitigating the risk of inappropriate interventions and adverse outcomes; and (ii) enabling targeted data correction strategies to improve overall data quality and ensure more reliable downstream analyses. While it is possible that our approach may produce potential false positives or negatives, the deviations it identifies are sufficiently informative to warrant clinical investigation prior to data ingestion into the EHR system. In this regard, the proposed approach functions as a practical and effective data fidelity screening mechanism, reinforcing the robustness and reliability of the EHR data infrastructure.
>
> > [a] Pandl, Konstantin D., et al. "Trustworthy machine learning for health care: scalable data valuation with the shapley value." Proceedings of the Conference on Health, Inference, and Learning. 2021.
>
> > [b] Bloch, Louise, Christoph M. Friedrich, and Alzheimer’s Disease Neuroimaging Initiative. "Data analysis with Shapley values for automatic subject selection in Alzheimer’s disease data sets using interpretable machine learning." Alzheimer's Research & Therapy 13.1 (2021): 155.

---

> ### Comment · Reviewer_oTHf · 2025-08-04
> **Official Comment by Reviewer oTHf**
>
> Thank you to the authors for the detailed responses. While some of my concerns have been addressed, the main issues I raised still remain unresolved.
>
> - First, the supplementary SE experiments verify only that permutation samplings produce consistent outputs; they do not quantify the deviation from the oracle. I am therefore still concerned that the error gap might widen after the two-stage distillation, given that earlier work employed permutation sampling in just one training stage with relatively simple ML models. In addition, RV was designed for time-series data, so its suitability here is unclear because the permutation samples are not time-series (please let me know if I have misunderstood).
>
> - Second, I remain unconvinced that domain shift can be disregarded when the framework is targeted to be evaluated at only a single institution or department. Domain shifts, especially temporal ones, frequently arise in real-world healthcare data and may manifest within a single department, let alone across the wider institution.
>
> - Third, the framework appears fragmented in real-world applications, as its performance seems highly sensitive to an ad-hoc threshold. It is unclear how a valid threshold can be predetermined before conducting any assessment, especially in the absence of a sensitivity analysis. Is there a general guideline for selecting this threshold, or must it always be determined/calibrated by clinicians on a case-by-case basis?
>
> I am willing to adjust my score if the authors can adequately address the above concerns.

---

> > ### Author Response · Authors · 2025-08-04
> >
> > We thank the reviewer for the follow-up and would like to provide further clarifications to address the remaining concerns.
> >
> > ### First Comment
> >
> > Our previous response directly addressed the original comment regarding the approximation error introduced by permutation sampling in lieu of exact Shapley value computation. As stated, exact computation is intractable due to exponential complexity, and permutation-based Monte Carlo estimation remains the feasible and widely accepted alternative in real-world applications [14, 46]. To assess the reliability of this approximation, we employed 100K independent samples with two convergence diagnostics: standard error (SE) and rolling variance (RV). These metrics demonstrated clear statistical convergence across all tasks, with SE dropping below 0.01% and RV approaching zero, indicating that further sampling yields negligible benefit. These results, together with established statistical theory, support that the approximation is sufficiently close to the true values for the purpose of training the neural oracle.
> >
> > We would like to clarify a possible misunderstanding about the RV metric. Although the term "rolling variance" is commonly used in time-series analysis, in our context RV is not applied to temporal data. Rather, it quantifies the local variance of the cumulative mean over a moving window of Monte Carlo estimates, i.e., it measures the temporal stability of the estimates as more samples are drawn. Each permutation sample is independent, and as the sample size increases, the estimate should stabilize. Thus, RV captures the expected reduction in local fluctuations of the estimator and serves as a practical signal of convergence, entirely unrelated to time-series modeling. We apologize for any confusion caused by the term.
> >
> > Further, regarding the comment that "earlier work" used permutation sampling in only one training stage with simpler models, we would appreciate clarification. If this refers to prior work that used permutation-based data Shapley value estimates to train relatively simple models in a single stage, we agree—this is indeed a common approach. Even with Monte Carlo estimation, computing reliable data Shapley values requires evaluating model outputs across numerous subsets of the data. When the model involved is complex, such computation becomes extremely costly, making convergence via permutation sampling practically unattainable in many cases. This explains why existing studies often limit themselves to simpler models or single-stage usage.
> >
> > Regarding the concern that "the error gap might widen after the two-stage distillation," we emphasize that only the first stage involves fitting a neural oracle to approximate the data Shapley values. The second stage trains a fidelity predictor $\Psi$ using the outputs of the neural oracle across multiple tasks. In the first stage, the Pearson correlation is around 0.95 between the data Shapley estimates and oracle predictions on held-out data, indicating high information preservation in the learned oracle. We also note that this oracle serves as a surrogate primarily for computational reasons and that similar oracle approximations have been used in prior work [6]. Importantly, our main contribution lies not in the data Shapley estimation itself, but in how this information is used for **detecting fidelity-related data deviations**. The experimental results demonstrate that the learned predictor is effective in improving clinical decision-making, despite the unavoidable approximations in the initial oracle construction.

---

> > ### Author Response · Authors · 2025-08-04
> >
> > ### Second Comment
> >
> > We understand that the reviewer’s concern may pertain to temporal domain shifts arising even within a single institution or department—for example, due to evolving clinical practice patterns, changing patient demographics, or updates in medical guidelines. If this is indeed the intended scope, we would like to emphasize that such shifts do not contradict the design or assumptions of our proposal. On the contrary, we view temporal evolution in healthcare data as a natural aspect of clinical workflows, much like the periodic revision of diagnostic criteria or treatment guidelines. Our approach is designed to support **pre-hoc detection of data deviations at the point of data entry**, and as such, it is expected—and indeed appropriate—for the underlying model to be periodically retrained to remain aligned with updated standards of care and population characteristics.
> >
> > To this end, our approach does not assume stationarity over long time horizons. Instead, it is structured to operate within a deployment lifecycle that allows model updates when meaningful shifts are detected. In practice, these updates can be scheduled in accordance with institutional retraining policies, or be triggered in response to systematic deviation alerts captured by our own data quality mechanisms.
> >
> > It is also possible that we have misunderstood the specific type of domain shift the reviewer had in mind. If the concern refers to some other form of temporal or latent sub-domain heterogeneity that we have not sufficiently addressed, we would greatly appreciate further clarification. Such guidance would help us better articulate limitations and more precisely scope our contributions.
> >
> > ### Third Comment
> >
> > We would like to clarify that from a methodological standpoint, the definition of deviation in our proposal is inherently tied to the sign of the change in $\Psi$, i.e., $\Delta \Psi$. In this theoretical formulation, the deviation threshold $\delta$ is set to zero, and the presence of deviation is signaled when $\Delta \Psi$ crosses zero. This makes our approach fundamentally insensitive to the choice of $\delta$, as the binary indicator is determined by the directionality rather than the magnitude of the change.
> >
> > Figure 4 in our paper supports this formulation empirically: it shows that across a range of controlled deviation injections, the sign of $\Delta \Psi$ reliably tracks the presence and direction of perturbations, even in near-boundary cases. Thus, the zero-threshold formulation already provides a valid and effective operationalization of deviation detection in controlled settings, without relying on any post-hoc parameter tuning.
> >
> > That said, we acknowledge that in real-world clinical deployments, additional considerations such as reducing false positives or aligning with institution-specific alerting protocols may motivate the use of a non-zero $\delta$ to introduce a margin of tolerance. In such cases, $\delta$ plays a role similar to that of alerting thresholds in clinical decision support systems—introduced not for theoretical necessity but for practical calibration. We agree that this calibration often requires input from domain experts, though it is grounded in retrospective analysis of nominal behavior (e.g., $\Psi$ distribution under non-deviated conditions) rather than being entirely ad hoc.
> >
> > While our main experiments focus on fixed-threshold and sign-based detection, we recognize the value of a more systematic analysis of threshold sensitivity and have planned such extensions for future deployment validations.
> >
> > We hope these additional clarifications address your concerns and stand ready for discussion and elaboration if further questions arise.

---

### Note · Authors · 2025-08-11

We thank the reviewers for their thoughtful engagement. Our work targets pre-hoc screening at the point of data entry. We introduce a multi-task, data-valuation-based EHR fidelity score ($\Psi$) and a practical two-stage distillation-and-detection workflow. Reviewers recognised the methodological novelty, clinical fit, and sensitivity to small-magnitude deviations; case studies and ablations support practical value. The majority opinion leans toward acceptance.

We have addressed every concern raised in the discussion with added analyses and clarifications. The data Shapley oracle approximation was validated by large-scale Monte Carlo estimates with tight errors and stability checks; the neural oracle preserves signal on held-out estimates. For thresholds, the theory uses the sign of $\Delta \Psi (\delta=0)$; in deployment, a small tolerance can be set from retrospective data with clinician input to manage alert load. On domain shifts, this study is single-institution and pre-hoc; risk is limited, and periodic retraining is part of the intended lifecycle. We also added targeted ablations and rule-based baselines and explained how our detector complements simple range checks. No open issue affects the core contribution.

We clarified the scope. The primary goal is to support real-time judgment at the point of data entry, a critical need in actual clinical practice, even though the approach can also be applied to post-hoc cleaning and to reducing noise in downstream applications.

We will improve clarity by framing inputs, outputs, and the controlled-deviation setup earlier, add concise guidance on threshold calibration, and expand the limitations to address temporal domain shifts and corruption generality. The introduction will be updated to balance the motivation from downstream EHR analytics with the immediate, clinician-validated need for pre-hoc detection.

In sum, all points raised in the exchange have been resolved. To our knowledge, this is the first approach that converts multi-task, model-based data valuation into a single deployable fidelity signal at data entry through a scalable two-stage distillation. This innovation strengthens the point-of-entry check, raising sensitivity beyond range-based rules while making pre-hoc screening feasible at scale. Large-scale validation, targeted ablations, and clinician-vetted cases support both the novelty and the practical value.

We appreciate your consideration.

---

### Decision · Program_Chairs · 2025-09-17

**Decision:**

Accept (poster)

**Comment:**

The paper tackles the important problem of detecting data deviation in electronic health records (EHRs) using a bi-level knowledge distillation approach. The authors define data deviation as discrepancies between recorded entries and a patient's actual physiological state, and propose a system that first approximates computationally expensive data Shapley values with neural oracles for individual clinical tasks, then distills these into a unified EHR data fidelity predictor. The approach aims to support pre-hoc detection of potential data quality issues at the point of data entry, enabling clinicians to identify and address erroneous records before they contaminate downstream clinical workflows.

The reviewers and authors had an active discussion with several substantive concerns. Reviewer oTHf raised critical questions about the reliability of using Monte Carlo approximation instead of exact Shapley values, the potential for domain shift issues, and the practical challenge of setting detection thresholds in clinical settings. The authors responded with extensive Monte Carlo validation studies showing convergence, but oTHf remained concerned about error propagation through the two-stage distillation process. Reviewer opND questioned the scope and practical utility of the approach, asking for better ablation studies and clarification about whether the goal was data cleaning or pre-hoc detection. The authors provided comprehensive ablation results and clarified their focus on pre-hoc screening. Reviewer 8yxT acknowledged the importance of the problem but questioned the generalizability of the perturbation strategy to real-world data entry errors and whether the method truly measures data fidelity versus feature impact on model performance. The authors maintained that their physiologically-scaled perturbations align with clinical quality control practices and that their multi-task approach provides a robust proxy for data fidelity.

In reviewing the paper, reviews, and author responses, I recommend acceptance because the paper addresses a clinically important and under-explored problem with a technically sound approach that demonstrates clear practical value. While legitimate concerns remain about the approximation errors in the Shapley value estimation and the distinction between measuring data utility versus true physiological fidelity, the authors provided substantial additional analyses addressing most reviewer concerns. The multi-task knowledge distillation framework is methodologically novel, the experimental validation is thorough with both controlled experiments and real-world case studies, and the clinical motivation is well-established. The work makes a meaningful contribution to healthcare AI by providing a principled approach to data quality assessment that could improve clinical decision-making, despite some remaining questions about generalizability and threshold calibration in practice.